# High-dimensional Contextual Bandit Problem without Sparsity

**Junpei Komiyama**
New York University
junpei@komiyama.info

**Masaaki Imaizumi**
The University of Tokyo / RIKEN Center for AIP
imaizumi@g.ecc.u-tokyo.ac.jp

## Abstract

In this research, we investigate the high-dimensional linear contextual bandit problem where the number of features $p$ is greater than the budget $T$, or it may even be infinite. Differing from the majority of previous works in this field, we do not impose sparsity on the regression coefficients. Instead, we rely on recent findings on overparameterized models, which enables us to analyze the performance of the minimum-norm interpolating estimator when data distributions have small effective ranks. We propose an explore-then-commit (EtC) algorithm to address this problem and examine its performance. Through our analysis, we derive the optimal rate of the ETC algorithm in terms of $T$ and show that this rate can be achieved by balancing exploration and exploitation. Moreover, we introduce an adaptive explore-then-commit (AEtC) algorithm that adaptively finds the optimal balance. We assess the performance of the proposed algorithms through a series of simulations.

## 1 Introduction

The multi-armed bandit problem [Robbins, 1952, Lai and Robbins, 1985] has been widely studied in the field of sequential decision-making problems in uncertain environments, and it can be applied to a variety of real-world scenarios. This problem involves an agent selecting one of $K$ arms in each round and receiving a corresponding reward. The agent aims to maximize the cumulative reward over rounds by using a clever algorithm that balances exploration and exploitation. In particular, a version of this problem called the contextual bandit problem [Abe and Long, 1999, Li et al., 2010] has attracted significant attention in the machine learning community. By observing the contexts associated with the arms, the agent can choose the best arm as a function of the contexts. This extension enables us to model many personalized machine learning scenarios, such as recommendation systems [Li et al., 2010, Wang et al., 2022] and online advertising [Tang et al., 2013], and personalized treatments [Chakraborty and Murphy, 2014].

Most of the papers about stochastic linear bandits assume that the number of features $p$ is moderate [Li et al., 2010, Chu et al., 2011, Abbasi-Yadkori et al., 2011]. When $p = o(\sqrt{T})$ to the number of rounds $T$, the model is identifiable, and the agent can choose the best arm for most rounds. However, recent machine learning models desire to utilize an even larger number of features, and the theory of bandit models under the identifiability assumption does not necessarily reflect the modern use of machine learning. Several recent papers have overcome this limitation by considering sparse linear bandit models [Wang et al., 2018, Kim and Paik, 2019, Bastani and Bayati, 2020, Hao et al., 2020, Oh et al., 2021, Li et al., 2022, Jang et al., 2022]. Sparse linear bandit models accept a very large number of features[1] and suppress most of the coefficients by introducing the $\ell 1$ regularize.

---

[1] Typically, the number of feature $p$ can be exponential to the number of datapoints $T$.

37th Conference on Neural Information Processing Systems (NeurIPS 2023).

That said, the sparsity imposed by such models limits the applicability of these models. For example, in the case of recommendation models based on factorization, each user is associated with a dense latent vector [Rendle, 2010, Agarwal et al., 2012, Wang et al., 2022], which implies the sparsity is not unlikely the case. Another possible drawback of sparse models is that it requires the condition number to be close to one (e.g., the restricted isometry property, see Van De Geer and Bühlmann [2009] for review). This implies that the quality of the estimator is compromised by the noise on the features that correspond to small eigenvalues. Furthermore, it is still non-trivial to select a proper value of the penalty coefficient as a hyper-parameter. For example, Hara and Maehara [2017] claims that small changes in the choice of coefficients significantly alter feature selection, and Miolane and Montanari [2021] show a limitation of the conventional theory on the choice of penalty coefficients.

In this paper, We consider an alternative high-dimensional linear bandit problem without sparsity. We allow $p$ to be as large as desired, and in fact, we even allow $p$ to be infinitely large. Such an overparameterized model has more parameters than the number of training data points. A natural estimator in such a case is an *interpolating* estimator, which perfectly fits the training data. We adopt recent results that bound the error of the estimator in terms of the *effective rank* [Koltchinskii and Lounici, 2017, Bartlett et al., 2020] on the covariance of the features. When the eigenvalues of the covariance decay moderately fast, we can obtain a concentration inequality on the squared error of the estimator.

The contributions of this paper are as follows: First, We consider explore-then-commit (EtC) strategy for the stochastic bandit problem based on the minimum-norm interpolating estimator. We derive the optimal rate of exploration that minimizes regret. However, EtC requires model-dependent parameters on the covariance, which limits the practical utility. To address this limitation, we propose an adaptive explore-then-commit (AEtC) strategy, which adaptively estimates these parameters and achieves the optimal rate. We conduct simulations to verify the efficiency of the proposed method.

## 2 Preliminary

### 2.1 Notation

For $z \in \mathbb{N}$, $[z] := \{1, 2, \ldots, z\}$. For $x \in \mathbb{R}$, the notation $\lfloor x \rfloor$ here denotes the largest integer that is less than or equal to a scalar $x$. For vectors $X, X' \in \mathbb{R}^p$, $\langle X, X' \rangle := X^\top X'$ is an inner product, $\|X\|_2^2 := \langle X, X \rangle$ is an $\ell 2$-norm. For a positive-definite matrix $A \in \mathbb{R}^p \times \mathbb{R}^p$, $\|X\|_A^2 := \langle X, AX \rangle$ is a weighted $\ell 2$-norm. $\|\mathbf{A}\|_{\mathrm{op}}$ denotes an operator norm of $\mathbf{A}$. $O(\cdot), o(\cdot), \Omega(\cdot), \omega(\cdot)$ and $\Theta(\cdot)$ denotes Landau's Big-O, little-o, Big-Omega, little-omega, and Big-Theta notations, respectively. $\widetilde{O}(\cdot), \widetilde{\Omega}(\cdot)$, and $\widetilde{\Theta}(\cdot)$ are the notations that ignore polylogarithmic factors.

### 2.2 Problem Setup

This paper considers a linear contextual bandit problem with $K$ arms. We consider the fully stochastic setting, where the contexts, as well as the rewards, are drawn from fixed distributions. For each round $t \in [T]$ and arm $i \in [K]$, we define $X^{(i)}(t)$ as a $p$-dimensional zero-mean sub-Gaussian vector. We assume $X^{(i)}(t)$ is independent among rounds (i.e., vectors in two different rounds $t, t'$ are independent) but allow vectors $X^{(1)}(t), X^{(2)}(t), \ldots, X^{(K)}(t)$ to be correlated with each other. The forecaster chooses an arm $I(t) \in [K]$ based on the $X^{(i)}(t)$ values of all the arms, and then observes a reward that follows a linear model as shown in

$$Y^{(I(t))}(t) = \langle X^{(I(t))}(t), \theta^{(I(t))} \rangle + \xi(t). \tag{1}$$

The unknown true parameters $\theta^{(i)}$ for each arm $i \in [K]$ lie in a parameter space $\Theta \subset \mathbb{R}^p$, and the independent noise term $\xi(t)$ has zero mean and variance $\sigma^2 > 0$. We assume that $\xi(t)$ is sub-Gaussian, and for the sake of simplicity, we assume that it does not depend on the choice of the arm. However, our results can be extended to the case where $\xi(t)$ varies among arms. We assume that each $\theta^{(i)}$ are bounded $\|\theta^{(i)}\|_2 \leq \theta_{\max}$. For each $i \in [K]$, we define a covariance matrix $\Sigma^{(i)} = \mathbb{E}[X^{(i)}(t) X^{(i)}(t)^\top] \in \mathbb{R}^{p \times p}$.

We define $i^*(t) := \mathrm{argmax}_{i \in [K]} \langle X^{(i)}(t), \theta^{(i)} \rangle$ as the (ex ante) optimal arm at round $t$. Our goal is to design an algorithm that maximizes the total reward, which is equivalent to minimizing the following

expected regret [Lai and Robbins, 1985, Auer et al., 2002];

$$R(T) := \mathbb{E}\left[\sum_{t=1}^{T}\left(\langle X^{(i^*(t))}(t), \theta^{(i^*(t))}\rangle - \langle X^{(I(t))}(t), \theta^{(I(t))}\rangle\right)\right], \tag{2}$$

where the expectation is taken with respect to the randomness of the contexts and (possibly) on the choice of arm $I(t)$.

## 2.3 Theory of Overparametrized Models

The primary focus of this paper is on scenarios where the number of arms $K$ is moderate, but the number of features $p$ is greater than the budget $T$, possibly to the point of being infinite. The efficiency of linear regression models in such scenarios depends significantly on the covariance matrix, $\Sigma^{(i)}$. Unlike sparsity, the theory of benign overfitting tightly examines errors using the decay of the eigenvalues of the covariance matrix of the context. In particular, if the decay rate of the eigenvalues is at a certain level, the error in linear regression converges to zero, even in high-dimensional spaces. To provide a more rigorous analysis of eigenvalue decay, the concept of *effective rank* is introduced in Section 2.4.

**Remark 1.** (Dependence on $T$) In accordance with Bartlett et al. [2020], we permit the covariance matrix $\Sigma^{(i)}$ to depend on $T$. In other words, we consider the sequence of covariances $\Sigma^{(i)}(1), \Sigma^{(i)}(2), \ldots$ for each $T = 1, 2, \ldots$. The linear regression is consistent if the effective rank of $\Sigma^{(i)}(T)$ grows sufficiently slow as a function of $T$.

## 2.4 Effective Ranks of Covariance Matrix

For a covariance matrix $\Sigma^{(i)}$ for $i \in [K]$ and $k \in [p]$, let $\lambda_k^{(i)}$ be its $k$-th largest eigenvalue, such that $\Sigma^{(i)} = \sum_{k=1}^{p} \lambda_k^{(i)} u_k^{(i)} (u_k^{(i)})^\top$ with order $\lambda_1^{(i)} \geq \lambda_2^{(i)} \geq \cdots \geq \lambda_p^{(i)}$ and eigenvectors $u_k^{(i)}$. We define the concept of *effective rank* as

$$r_k(\Sigma^{(i)}) := \frac{\sum_{j>k} \lambda_j^{(i)}}{\lambda_{k+1}^{(i)}}, \quad \text{and} \quad R_k(\Sigma^{(i)}) := \frac{(\sum_{j>k} \lambda_j^{(i)})^2}{\sum_{j>k} (\lambda_j^{(i)})^2}. \tag{3}$$

The first quantity $r_k(\Sigma^{(i)})$ is related to the trace of $\Sigma^{(i)}$, and the second quantity $R_k(\Sigma^{(i)})$ is related to the decay rate of the eigenvalues.

The effective rank is used as a measure of the intrinsic complexity of high-dimensional data, by rigorously capturing a decay rate of eigenvalues. If the covariance matrix $\Sigma^{(i)}$ is an identity matrix of size $p$, then $r_0(\Sigma^{(i)})$ and $R_0(\Sigma^{(i)})$ are both equal to $p$ (the rank of $\Sigma^{(i)}$). However, we anticipate that these quantities will be less than $p$, which enables learning with fewer samples. In Figure 1, we plot of the effective rank in $k$ with certain cases of eigenvalues $\{\lambda_k\}_k$ of $\Sigma$. Even though $\Sigma$ is full-rank, the effective rank $R_k(\Sigma)$ is sublinear in $k$, which reflects the intrinsic low-complexity of data.

Because of this property, the effective ranks are used in modern high-dimensional statistics. Koltchinskii and Lounici [2017] evaluated concentration inequalities for high-dimensional random matrices using the notion. Bartlett et al. [2020], Tsigler and Bartlett [2020] evaluated the prediction error of a high-dimensional linear regression using the effective ranks and showed the consistency of the prediction.

## 3 Explore-then-Commit with Interpolation

The *Explore-then-Commit* (EtC) algorithm is a well-known approach for solving high-dimensional linear bandit problems, and it has been shown to be effective in previous studies such as Hao et al. [2020], Li et al. [2022]. The EtC algorithm operates by first conducting $T_0 = NK < T$ rounds of exploration, during which it uniformly explores all available arms to construct an estimator $\widehat{\theta}^{(i)}$ for each arm $i \in [K]$. After the exploration phase, the algorithm proceeds with exploitation. Let $N$ be the number of the draws of arm $i$, and $\mathbf{X}^{(i)} = (X_1^{(i)}, ..., X_N^{(i)})^\top \in \mathbb{R}^{N \times p}$ and $\mathbf{Y}^{(i)} = (Y_1^{(i)}, ..., Y_N^{(i)})^\top \in \mathbb{R}^N$

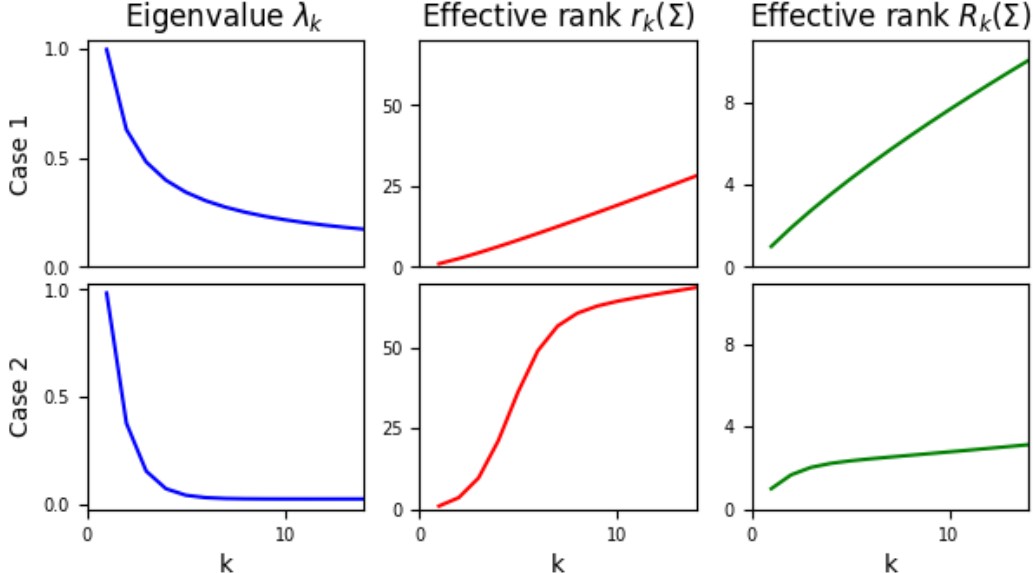

Figure 1: Eigenvalues $\lambda_k$ (left), the effective rank $r_k(\Sigma)$ (middle), and the effective rank $R_k(\Sigma)$ (right) with $k = 1, ..., 15$ and $T = 3$; Case 1: $\lambda_k = Ck^{-1+1/T^{0.99}}$, and Case 2: $\lambda_k = C(\exp(-k) + T\exp(-T)/p)$. Case 1 considers a decay that is slightly faster than $k^{-1}$, whereas Case 2 considers an exponentially fast decay. The slower increase of $r_k(\Sigma)$ and $R_k(\Sigma)$ in Case 2 reflects the impact of the eigenvalues' faster decay.

be the observed contexts and rewards of arm $i$, where $(X_n^{(i)}, Y_n^{(i)})$ is the corresponding values on the $n$-th draw of arm $i$. Since we choose $I(t)$ uniformly during the exploration phase, these are independent and identically drawn samples from the corresponding distribution.

For estimating the parameter $\theta^{(i)}$, we consider the minimum-norm *interpolating estimator* that perfectly fits the data, which reveals its advantage in recent results on high-dimensions [Bartlett et al., 2020]. Rigorously, we assume $p > N$ and and consider the following definition:

$$\widehat{\theta}^{(i)} := \operatorname{argmin}\left\{\|\theta\|_2 \,\middle|\, \theta \in \mathbb{R}^p, 0 = \sum_{(Y_n^{(i)}, X_n^{(i)}):n \leq N} (Y_n^{(i)} - \langle \theta, X_n^{(i)} \rangle)^2\right\}.$$

This estimator has a simple representation $\widehat{\theta}^{(i)} = (\mathbf{X}^{(i)})^\top (\mathbf{X}^{(i)}(\mathbf{X}^{(i)})^\top)^{-1}\mathbf{Y}^{(i)}$. The EtC algorithm is presented in Algorithm 1, which utilizes the aforementioned interpolating estimator. In the subsequent sections, we will first discuss the assumptions on the data-generating process, followed by an analysis of the accuracy of the estimator. We then present an upper bounds on the regret of the EtC algorithm.

## 4 Theory of Explore-then-Commit

This section analyzes the EtC algorithm. If we were to fix the number of samples, this theory would largely align with the existing literature [Bartlett et al., 2020, Tsigler and Bartlett, 2020]. However, the unique challenge in our case arises from the fact that the EtC selects the number of samples used to construct an estimator to balance between exploration and exploitation.

### 4.1 Assumption and Notion

We consider a spectral decomposition $\Sigma^{(i)} = V^{(i)}\Lambda^{(i)}(V^{(i)})^\top$ with the matrices (operators) $V^{(i)}$ and $\Lambda^{(i)}$ for each arm $i \in [K]$, then impose the following assumptions.

**Assumption 1** (sub-Gaussianity). *For all $t \in \mathbb{N}$ and $i \in [K]$, the followings hold:*

---

**Algorithm 1** Explore-then-Commit (EtC)

---

**Require:** Exploration duration $T_0$.
  **for** $t = 1, .., T_0$ **do**
    Observe $X^{(i)}(t)$ for all $i \in [K]$.
    Choose $I(t) = t - K\lfloor t/K \rfloor$.
    Receive a reward $Y^{(I(t))}(t)$.
  **end for**
  **for** $i \in [K]$ **do**
    $\widehat{\theta}^{(i)} \leftarrow (\mathbf{X}^{(i)})^\top (\mathbf{X}^{(i)}(\mathbf{X}^{(i)})^\top)^{-1} \mathbf{Y}^{(i)}$.
  **end for**
  **for** $t = T_0 + 1, ..., T$ **do**
    Observe $X^{(i)}(t)$ for all $i \in [K]$.
    Choose $I(t) = \mathrm{argmax}_{i \in [K]} \langle X^{(i)}(t), \widehat{\theta}^{(i)} \rangle$.
    Receive a reward $Y^{(I(t))}(t)$.
  **end for**

---

- *There exists a random vector $Z^{(i)}(t)$ such that $X^{(i)}(t) = V^{(i)}(\Lambda^{(i)})^{1/2}Z^{(i)}(t)$ which is independent elements and sub-Gaussian with a parameter $\kappa_x^2 > 0$, that is, for all $\lambda \in \mathbb{R}^p$, we have $\mathbb{E}[\exp(\langle \lambda, Z^{(i)}(t) \rangle)] \le \exp(\kappa_x^2 \|\lambda\|_2^2 / 2)$.*

- *Moreover, $\xi(t)$ is conditionally sub-Gaussian with a parameter $\kappa_\xi^2 > 0$, that is, for all $\lambda \in \mathbb{R}$, we have $\mathbb{E}[\exp(\lambda \xi(t)) \mid X^{(i)}(t)] \le \exp(\kappa_\xi^2 \lambda^2 / 2)$.*

The first bullet point of Assumption 1 requires that the features are multivariate sub-Gaussian and their tail is not too heavy, which is important for concentration inequalities on random variables [Vershynin, 2018]. The second bullet point of Assumption 1, which states that rewards involve sub-Gaussian noise, is a common assumption in contextual bandit problems.

## 4.2 Benign Covariance

We impose a condition to be benign on the covariance matrix $\Sigma^{(i)}$ for all $i \in [K]$. The condition is described using the notion of effective/coherent rank, which is commonly applied to the study of benign overfitting [Bartlett et al., 2020, Tsigler and Bartlett, 2020]. However, unlike those papers that estimate using all $T$ samples, we only use a portion of the data for learning.

We first define the *coherent rank* of $\Sigma^{(i)}$ with a number of samples $N$ as

$$k_N^* = k_N^*(\Sigma^{(i)}, N) := \min\{k \in \mathbb{N} \cup \{0\} \mid r_k(\Sigma^{(i)}) \ge N\},$$

where we define $\min\{\emptyset\} = +\infty$. By using the coherent rank, we decompose the error into two quantities called *effective bias/variance* denoted as $B_{N,T}^{(i)}$ and $V_{N,T}^{(i)}$, based on a budget of $T$ and the number of samples $N$ used for estimation.

$$B_{N,T}^{(i)} := \lambda_{k_N^*}^{(i)}, \text{ and } V_{N,T}^{(i)} := \left( \frac{k_N^*}{N} + \frac{N}{R_{k_N^*}(\Sigma^{(i)})} \right). \tag{4}$$

**Definition 1** (Benign covariance). Under Assumption 1, a covariance matrix $\Sigma^{(i)}$ is *benign*, if $B_{N,T}^{(i)} = o(1)$ and $V_{N,T}^{(i)} = o(1)$ hold as $N, T \to \infty$ while $N = \Theta(T)$.

To put it differently, Assumption 1 and Definition 1 state that, if we can achieve consistent estimation by using all the data for estimation, then the covariance matrix is considered benign.

Technically, the benign property implies that eigenvalues decay fast enough compared with $T$. In particular, the following examples have been considered in the literature.

**Proposition 1** (Example of Benign Covariance, Theorem 31 in Bartlett et al. [2020][2] ). *A covariance matrix $\Sigma^{(i)}$ with eigenvalues $\lambda_1^{(i)}, \lambda_2^{(i)}, \dots$ is benign if it satisfies one of the following.*

---

[2]It should be noted that Bartlett et al. [2020] only provided the variance term. Later on, Tsigler and Bartlett [2020] described both the variance and bias terms, which we follow.

- **Example 1:** Let $a \in (0, 1)$, $p = \infty$ and $\lambda_k^{(i)} = k^{-(1+1/T^a)}$. In this case, we have $B_{N,T}^{(i)} = O((T^a/N)^{1+1/T^a})$ and $V_{N,T}^{(i)} = O(T^a/N + T^{-a})$. For this model to be learnable, $N \gg T^a$ is required, and the variance dominates the bias.

- **Example 2:** Let $\lambda_k^{(i)} = k^{-b}$ and $p = p_T = O(T^c)$ with $b \in (0, 1)$ and $c \in \left( \max\left(1, \frac{2}{2-b}, \frac{1}{1-b^2}\right), \frac{1}{1-b} \right)$. In this case, we have $B_{N,T}^{(i)} = O(T^{c(1-b)}/N)$ and $V_{N,T}^{(i)} = O((N/T^c)^{\max(1, \frac{1-b}{b})})$. For this model to be learnable, $N \gg T^{c(1-b)}$ is required, and the bias dominates the variance.

The first example is when the decay rate is near $k^{-1}$, but the trace is $O(T^a)$. The second example is when the decay rate is smaller than $k^{-1}$, and the trace is $O(T^{c(1-b)})$. Note that Bartlett et al. [2020] provided two other examples of the benign covariance matrices.[3] Our analysis mainly focuses on the two examples in Proposition 1, but another example is also empirically tested in Section 6.

### 4.3 Estimation Error by Exploration

We evaluate an error in the estimator $\widehat{\theta}^{(i)}$ by its prediction quality. That is, with a covariance matrix $\Sigma^{(i)}$ and an identical copy $X_*^{(i)}$ of $X_1^{(i)}$, we study

$$\|\theta^{(i)} - \widehat{\theta}^{(i)}\|_{\Sigma^{(i)}}^2 = \mathbb{E}_{X_*^{(i)}}\left[ (\langle \theta^{(i)}, X_*^{(i)} \rangle - \langle \widehat{\theta}^{(i)}, X_*^{(i)} \rangle)^2 \right],$$

The following result bounds the error of the estimator $\widehat{\theta}^{(i)}$ in terms of bias and variance, which is a slight extension of Tsigler and Bartlett [2020]. For $k \in [p]$, we define an empirical submatrix as $\mathbf{X}_{k:\infty}^{(i)} \in \mathbb{R}^{N \times (p-k)}$ as the $p - k$ columns to the right of $\mathbf{X}^{(i)}$, and define a Gram sub-matrix $A_k^{(i)} = \mathbf{X}_{k:\infty}^{(i)}(\mathbf{X}_{k:\infty}^{(i)})^\top \in \mathbb{R}^{N \times N}$.

**Theorem 2.** *Suppose Assumption 1. If there exists $c_U > 1$ such that $k_N^* < N/c_U$ and a condition number of $A_k^{(i)}$ is positive with probability at least $1 - c_U e^{-N/c_U}$, then we have*

$$\|\widehat{\theta}^{(i)} - \theta^{(i)}\|_{\Sigma^{(i)}}^2 \leq C_U \left( B_{N,T}^{(i)} + V_{N,T}^{(i)} \right), \tag{5}$$

*with some constant $C_U > 0$ and probability at least $1 - 2c_U e^{-N/c_U}$.*

Theorem 2 implies that the estimation error converges to zero if $\Sigma^{(i)}$ has the benign property. The assumption on the condition number of $A_k^{(i)}$ has a sufficient condition, which is provided in Lemma 3 in Tsigler and Bartlett [2020].

Moreover, the following lemma implies the tightness of the analysis in Theorem 2.

**Lemma 3** (Lower bound of estimation error, Theorem 10 in Tsigler and Bartlett [2020]). *Suppose Assumption 1. These exist some constants $c_L, C_L > 0$ such that, with probability at least $1 - c_L e^{-N/c_L}$, we have*

$$\|\widehat{\theta}^{(i)} - \theta^{(i)}\|_{\Sigma^{(i)}}^2 \geq C_L \left( B_{N,T}^{(i)} + V_{N,T}^{(i)} \right). \tag{6}$$

In other words, the upper bound in Theorem 2 is optimal up to a constant.[4]

### 4.4 Regret Bound of Explore-then-Commit

This section analyzes the EtC algorithm. We introduce an error function $\mathrm{E}(N, T)$ that characterizes the rate of regret, which can be obtained by considering $(B_{N,T}^{(i)} + V_{N,T}^{(i)})^{1/2}$.

**Assumption 2.** (Error function[5]) *There exists a continuous function $\mathrm{E} : \mathbb{R}^2 \to \mathbb{R}^+$ that is decreasing in $N$ such that $\|\widehat{\theta}^{(i)} - \theta^{(i)}\|_{\Sigma^{(i)}} = \widetilde{\Theta}(\mathrm{E}(N, T))$ as $N, T \to \infty$.*

---

[3]One of the omitted examples is the case where eigenvalues decay slightly slower than Example 1. The other example is the case where eigenvalues decay exponentially but has a noise term.

[4]Note that the constant here can depend on model parameters.

[5]The coherent rank $k_N^*$ as well as $B_{N,T}^{(i)}, V_{N,T}^{(i)}$ are discrete in $N, T$, and the error function $\mathrm{E}(N, T)$ is introduced to circumvent the issues related to this discreteness.

Assumption 2 is satisfied in Examples 1 and 2. In particular, for Example 1 in Proposition 1, the error function is given as $E(N, T) = \sqrt{T^a/N + T^{-a}}$, while for Example 2 in Proposition 1, it is given as $E(N, T) = \sqrt{T^{c(1-b)}/N}$.

**Theorem 4.** *Suppose Assumptions 1–2. Suppose that we run the EtC algorithm (Algorithm 1). Then, regret (2) satisfies*

$$R(T) = \widetilde{O}(L(T, K)),$$

*as $T \to \infty$ with some $\alpha > 0$. where $L(T, K)$ is such that*

$$N^* = \min_N \{N : N \geq T \, E(N/K, T)\}. \tag{7}$$

*Specifically, for Example 1 in Proposition 1, we have $L(T, K) = \max(T^{(2+a)/3}K^{2/3}, T^{1-a/2})$, whereas for Example 2 in Proposition 1, we have $L(T, K) = T^{(2+c(1-b))/3}K^{2/3}$.*

*Proof sketch of Theorem 4.* We show that the regret-per-round is $O(1)$ during the exploration phase. Moreover, regret-per-round is $\widetilde{O}(E(T_0/K, T))$ during the exploitation phase. The total regret is $T_0 + \widetilde{O}(E(T_0/K, T))T$, and optimizing $T_0$ by using the decreasing property of $E(N, T)$ in $N$ yields the desired result. □

## 4.5 Matching Lower Bound

We show that this choice of $T_0$ as a function of $T$ is indeed optimal.

**Theorem 5.** *Suppose Assumptions 1–2. Assume that we run the EtC algorithm (Algorithm 1). For any choice of $T_0$, the following event occurs with strictly positive probability as $T \to \infty$:*

$$R(T) = \widetilde{\Omega}\left(L(T, 3)\right).$$

*Proof sketch of Theorem 5.* We explicitly construct a model with $K = 3$. Let $\epsilon_T = E(T_0/K, T)$. In the model, the $\Sigma^{(1)}, \Sigma^{(2)}, \Sigma^{(3)}$ are identical, and the only difference is that the first coefficients $\theta_1^{(1)} = 1, \theta_1^{(2)} = \Theta(\epsilon_T), \theta_1^{(3)} = 0$. All other coefficients are set to zero. Roughly speaking, the gap is

$$\left| \langle X^{(i)}(t), \widehat{\theta}^{(i)} \rangle - \langle X^{(j)}(t), \widehat{\theta}^{(j)} \rangle \right| = \begin{cases} \Theta(1) & (i = 1, j = 2, 3) \\ \Theta(\epsilon_T) & (i = 2, j = 3) \end{cases} .$$

The regret-per-round during the exploration phase is $\Theta(1)$ due to a misidentification of the best arm between arm 1 and arms $\{2, 3\}$. The regret-per-round during the exploitation phase is $\Theta(\epsilon_T)$ due to a misidentification of the best arm between arm 2 and arm 3. □

# 5 Adaptive Explore-then-Commit (AEtC) Algorithm

In the prior section, we demonstrated that the optimal way to minimize EtC's regret is by selecting $T_0$, with $T_0$ balancing the exploration and the exploitation. However, this requires knowledge of the covariance matrix's spectrum, which can be difficult to obtain in advance in certain scenarios. This section explores the way to adaptively determines the extent of exploration required.

## 5.1 Estimator

We assume that $\Sigma^{(i)}$ follows the data-generating process of Example 1 or 2 in Proposition 1. We use $\beta_T$ to denote that

$$\lambda_j^{(i)} = C_\lambda j^{-\beta_T}, \tag{8}$$

with some constant $C_\lambda > 0$. We have $\beta_T = 1 + 1/T^a$ for Example 1, and $\beta_T = b(< 1)$ for Example 2.

Balancing exploration and exploitation in an overparameterized model is challenging for the following reasons. First, the number of features $p$ is very large or even infinite[6]. Second, the trace is heavy-tail

---

[6]Namely, $p = \infty$ for for Proposition 1 (1) or $p = T^c$ in for Proposition 1 (2).

because the decay is not very fast. As a result, a naive use of a traditional method does not work. We address this issue by utilizing two estimators. The first estimator is on the trace $\text{tr}(\widehat{\Sigma}^{(i)})$ that we extracted from overparameterization theory [Bartlett et al., 2020], whereas the second estimator is on the estimated decay rate $\widehat{\beta}_T$ that derives from Bosq [2000], Koltchinskii and Lounici [2017].

For an arm $i \in [K]$ with $N = T_0/K$ draws, we define an estimated eigenvalues $\widehat{\lambda}_1^{(i)}, ..., \widehat{\lambda}_N^{(i)} > 0$ from an empirical covariance matrix $\widehat{\Sigma}^{(i)} := N^{-1} \sum_{j=1}^N X_j^{(i)}(t) X_j^{(i)}(t)^\top$. We define the empirical trace to be $\text{tr}(\widehat{\Sigma}^{(i)}) = \sum_j \widehat{\lambda}_j^{(i)}$. The following lemma states the consistency of the estimated trace under very mild conditions.

**Lemma 6.** (Error bound of empirical trace) *Suppose Assumption 1. Suppose the data generating process (DGP) of Example 1 or Example 2 in Proposition 1. Then, for any $C_{\text{poly}} > 0$, with probability at least $1 - 1/T^{C_{\text{poly}}}$, we have the following as $T \to \infty$:*

$$\frac{|\text{tr}(\Sigma^{(i)}) - \text{tr}(\widehat{\Sigma}^{(i)})|}{\text{tr}(\Sigma^{(i)})} = \widetilde{O}\left(\frac{\sqrt{\sum_j (\lambda_j^{(i)})^2}}{\sum_j \lambda_j^{(i)}}\right) = o(1). \tag{9}$$

Moreover, the following bounds the estimation error of each eigenvalue.

**Lemma 7.** *Suppose Assumption 1. For any $\delta \in (0, 1)$, for any $C_{\text{poly}} > 0$, with probability at least $1 - 1/T^{C_{\text{poly}}}$, we have the following for any $j = 1, ..., p$ as $T \to \infty$:*

$$\left|\widehat{\lambda}_j^{(i)} - \lambda_j^{(i)}\right| = O\left(\sqrt{\frac{\text{tr}(\Sigma^{(i)})}{N}}\right). \tag{10}$$

Lemma 7 implies the estimator of each eigenvalue is $o(1)$ if we choose $N \gg \text{tr}(\Sigma^{(i)})$. However, even if we choose a large $N$, the error is still non-negligible for the tail of eigenvalues where $|\widehat{\lambda}_j^{(i)} - \lambda_j^{(i)}|$ is very small[7]. Consequently, $\widehat{\lambda}_j^{(i)}$ for large $j$ are not necessarily consistent, so as to the effective rank and the coherent rank estimated from them. To address this, we estimate the decay rate $\beta_T$, and then estimate the subsequent statistics.

Let the estimated decay rate be $\widehat{\beta}_T = (1/\tau) \sum_{k=1}^\tau \log(\widehat{\lambda}_k / \widehat{\lambda}_{k+1}) / \log((k+1)/k)$. Theoretically, $\widehat{\beta}_T$ is consistent for any constant $\tau > 1$. In practice, we can use a reasonable constant $\tau$, such as $\tau = 10$, for robustness. To estimate the effective ranks, we use the following form

$$\widetilde{\lambda}_k^{(i)} = \widehat{\lambda}_1^{(i)} k^{-\widehat{\beta}_T}.$$

Namely, we consider empirical analogues of the effective rank for $k$ as

$$\widehat{r}_k(\Sigma^{(i)}) := \frac{\text{tr}(\widehat{\Sigma}^{(i)})}{\widetilde{\lambda}_{k+1}^{(i)}}, \text{ and } \widehat{R}_k(\Sigma^{(i)}) := \frac{(\text{tr}(\widehat{\Sigma}^{(i)}))^2}{\sum_{j>k} (\widetilde{\lambda}_j^{(i)})^2}.$$

We also define an estimator for the coherent rank as $\widehat{k}_N := \min\{k \in \mathbb{N} \cup \{0\} \mid \widehat{r}_k(\Sigma^{(i)}) \geq N\}$. Further, we define estimators of $B_{N,T}^{(i)}, V_{N,T}^{(i)}$ of Eq. (4) as

$$\widehat{B}_{N,T}^{(i)} := \widetilde{\lambda}_{\widehat{k}_N}^{(i)}, \text{ and } \widehat{V}_{N,T}^{(i)} := \left(\frac{\widehat{k}_N}{N} + \frac{N}{\widehat{R}_{\widehat{k}_N}(\Sigma^{(i)})}\right),$$

Note that the estimators $\widehat{r}_k$ and $\widehat{R}_k$ use the trace $\text{tr}(\widehat{\Sigma}^{(i)})$, which is a sum of eigenvalues, instead of the partial sum of the eigenvalues that constitutes the effective rank of Eq. (3). Despite this change, this does not affect[8] asymptotic consistency of the coherent rank estimator $\widehat{k}_N$.

The following lemma states that small $\tau$ suffices to assure the quality of $\widehat{\beta}^{(i)}$. We study a convergence rate of $\widehat{\beta}^{(i)}$ and the other estimators as follows.

---

[7]Remember that we consider $\Sigma^{(i)} = \Sigma^{(i)}(T)$ that depends on $T$. Tail of eigenvalues are $o(1)$ to $T$ as well.
[8]This is because $\sum_j j^{-\beta_T}$ with $\beta_T < 1$ or $\beta_T \approx 1$ is tail-heavy.

---

**Algorithm 2** Adaptive Explore-then-Commit (AEtC)

---
**while** $t = 1, 2, 3, ...$ **do**
    Observe $X^{(i)}(t)$ for all $i \in [K]$.
    Choose $I(t) = t - K\lfloor t/K \rfloor$.
    Receive a reward $Y^{(I(t))}(t)$.
    **if** $t \in \{\lfloor e^N \rfloor \mid N \in \mathbb{N}, N \geq \log T\}$ **then**
        **if** $\text{Stop}(t/K)$ **then**
            $T_0 \leftarrow t$.
            Break.
        **end if**
    **end if**
**end while**
**for** $i \in [K]$ **do**
    $\widehat{\theta}^{(i)} \leftarrow (\mathbf{X}^{(i)})^\top (\mathbf{X}^{(i)} (\mathbf{X}^{(i)})^\top)^{-1} \mathbf{Y}^{(i)}$.
**end for**
**for** $t = T_0 + 1, ..., T$ **do**
    Observe $X^{(i)}(t)$ for all $i \in [K]$.
    Choose $I(t) = \text{argmax}_{i \in [K]} \langle X^{(i)}(t), \widehat{\theta}^{(i)} \rangle$.
    Receive a reward $Y^{(I(t))}(t)$.
**end for**

---

**Lemma 8.** *Suppose Assumption 1. Suppose DGP of Example 1 or Example 2 in Proposition 1.[9] Assume that we choose $N = N(T)$ such that $\text{tr}(\Sigma^{(i)})/N = o(1)$ for all $i$. Then, for any $C_{\text{poly}} > 0$, with probability at least $1 - 1/T^{C_{\text{poly}}}$, we have*

$$\frac{|\widetilde{\lambda}_k^{(i)} - \lambda_k^{(i)}|}{\lambda_k^{(i)}} = o(1),$$

*as $T \to \infty$ for all $k \in [\tau]$. Moreover, it implies $|\beta_T - \widehat{\beta}_T| = o(1)$ and*

$$\max\left\{ \frac{\widehat{B}_{N,T}^{(i)}}{B_{N,T}^{(i)}}, \frac{\widehat{V}_{N,T}^{(i)}}{V_{N,T}^{(i)}} \right\} = T^{o(1)}. \tag{11}$$

Eq. (11) states that the estimated rate of error is accurate as $T \to \infty$. To see this, for Example 1 in Proposition 1, the estimated error rate is $T^{o(1)}(\sqrt{T^a/N + T^{-a}})$, whose exponent approaches to $\sqrt{T^a/N + T^{-a}}$ as $T \to \infty$.

## 5.2 Adaptive Explore-then-Commit (AEtC)

The Adaptive Explore-then-Commit (AEtC) algorithm is described in Algorithm 2. Unlike EtC, the amount of exploration in AEtC is adaptively determined based on the stopping condition:

$$\text{Stop}^{(i)}(N) = \left\{ N > C_T \text{tr}(\widehat{\Sigma}^{(i)}) \cap NK \geq T\sqrt{\widehat{B}_{N,T}^{(i)} + \widehat{V}_{N,T}^{(i)}} \right\},$$

and $\text{Stop} = \cap_{i \in [K]} \text{Stop}^{(i)}$, where $C_T = C_T(T)$ is a function of $T$ that slowly diverges to $+\infty$. The first condition $N > C_T \text{tr}(\widehat{\Sigma}^{(i)})$ ensures that $N$ is large enough to have consistency on the estimators, whereas the second condition $NK \geq T\sqrt{\widehat{B}_{N,T}^{(i)} + \widehat{V}_{N,T}^{(i)}}$ balances the exploration and exploitation. The following theorem bounds the regret of AEtC.

**Theorem 9.** *Suppose Assumption 1. Suppose DGP of Example 1 or Example 2 in Proposition 1. Then, for any $c > 0$, the regret (2) when we run the AEtC algorithm (Algorithm 2) with a sufficiently slowly diverging $C_T$ is bounded as follows as $T \to \infty$:*

$$R(T) = \widetilde{O}(L(T, K)T^c),$$

*where $L(T, K)$ is the function defined in Theorem 4.*

---
[9]Note that DGP of Example 1 or 2 implies the existence of the error function of Assumption 2.

Theorem 9 states that, we can choose arbitrarily small $c > 0$ and the exponent of the regret of AEtC approaches that of EtC (Theorem 4). Note that the condition $N \geq C_T \text{tr}(\widehat{\Sigma}^{(i)})$ should not dominate the balance between exploration and exploitation: For example, $C_T = C \log T$ for any $C > 0$ suffices. In practice, a constant $C_T$ works.

*Proof sketch of Theorem 9.* We can derive that Eqs. (9), (10), and (11) for all $N \in [T]$ that hold with probability at least $1 - O(1/T)$ by setting $C_{\text{poly}} = 2$ and taking a union bound over $N$, and thus

$$\bigcap_{i \in [K]} \bigcap_{N \geq C_T \text{tr}(\widehat{\Sigma}^{(i)})} \left\{ \log \left( \frac{\sqrt{\widehat{B}_{N,T}^{(i)} + \widehat{V}_{N,T}^{(i)}}}{\sqrt{B_{N,T}^{(i)} + V_{N,T}^{(i)}}} \right) = \log(T^{o(1)}) \right\} \tag{12}$$

holds. Under this event, the stopping time of AEtC and EtC are at most $T^{o(1)}$ times different. Given this, the proof of the theorem is easily modified from the proof of Theorem 4. □

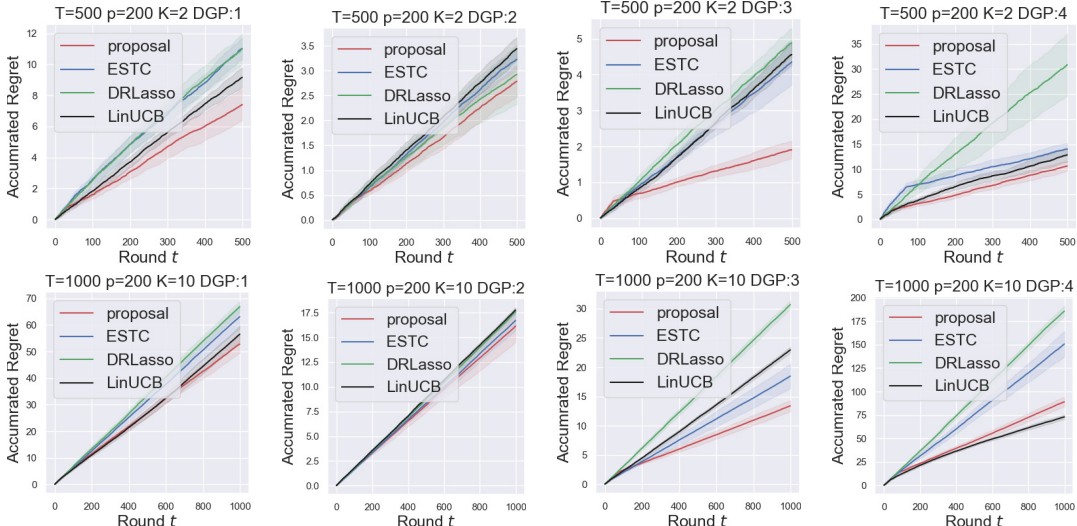

Figure 2: Comparison of methods on four DGPs. Smaller regret indicates better performance. The solid lines are the mean of 10 repetitions and the bands represent the standard deviation.

## 6 Simulation

We consider two setups: $(K, p, T) = (2, 200, 500)$ and $(K, p, T) = (10, 200, 1000)$. For each setup, we consider a covariance matrix $\Sigma^{(i)} = c^{(i)} \bar{\Sigma}$ where $c^{(i)} \sim \text{Uni}([0.5, 1.5])$ and a base covariance $\bar{\Sigma}$ for each $i \in [K]$, which represents a heterogeneous covariance among the arms. The base covariance $\bar{\Sigma}$ follows the following configurations: DGP 1: $\bar{\Sigma} = \text{diag}(\lambda_1, ..., \lambda_p), \lambda_k = k^{-0.5}$, DGP 2: $\bar{\Sigma} = \text{diag}(\lambda_1, ..., \lambda_p), \lambda_k = \exp(-k) + T \exp(-T)/p$, DGP 3: $\bar{\Sigma} = \text{diag}(\lambda_1, ..., \lambda_p), \lambda_k = k^{-1+1/T}$, and DGP 4: $\bar{\Sigma}_{i,j} = 0.3$ and $\bar{\Sigma}_{i,i} = 0.7$ for $j \neq i \in [p]$. Note that the DGPs 1 and 3 correspond to Examples 2 and 1 in Proposition 1. DGP 3 is benign as well, while DGP 4 is not. We generate $X^{(i)}(t)$ from a centered $p$-dimensional Gaussian with covariance $\Sigma^{(i)}$, $\theta^{(i)}$ from a standard normal distribution which yields non-sparse parameter vectors, and $Y^{(i)}(t)$ by the linear model (1) with the noise with variance $\sigma^2 = 0.01$. We compare proposal (AEtC, $C_T = 2$) to ESTC [Hao et al., 2020], LinUCB [Li et al., 2010, Abbasi-Yadkori et al., 2011], and DR Lasso Bandit [Kim and Paik, 2019].

Figure 2 illustrates the accumulated regret $R(t)$ in relation to the round $t \in [T]$. In the first three benign DGPs, the AEtC consistently outperforms the other methods. In DGP 2, the advantage of AEtC is insignificant because the regret of any method is already small. This is because in DGP 2, a model with exponentially decaying behavior, only a small fraction of eigenvalues are important. However, AEtC performs significantly better than its competitors in DGP 1 and DGP 3, where the eigenvalues have a heavy-tail distribution. In a non-benign example of DGP 4, the proposed method is still comparable to LinUCB, which demonstrates the utility of an interpolating estimator.

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
