# A    Limitations

The following characterizes the limitations. We consider addressing these as interesting directions for future work.

- **More adaptive algorithms, such as ones based on upper confidence bounds:** This paper considers the class of the explore-then-commit methods. In many bandit problems, the upper confidence bound (UCB) method provides better empirical performance since it adaptively balances exploration and exploitation. Applying UCB to this problem is important future work. The key challenge here is that such an adaptive estimator involves a biased selection of the context vectors, which requires a more adaptive error bound for a high-dimensional linear model, such as the self-normalized bound [Abbasi-Yadkori et al., 2011].

- **Lower bound of the problem:** While we showed the optimal choice of $T_0$ by deriving a matching bound, this does not exclude the possibility of a more adaptive bandit algorithm (e.g., UCB) that achieves a better rate of regret. Explicit construction of the lower bound requires two different models where the probability of an accurate (low-regret) estimate in one model implies a misestimation in the other model. To our knowledge, such a process for our non-sparse high-dimensional model is challenging because, unlike the sparse bandit model where only a small amount of parameters are active, in the non-sparse regime, we need to devise two models where very large ($p > n$) number of non-equal coefficients and need to bound the KL divergence between such a large multivariate Gaussians.

- **Temporal correlation:** This paper assumes temporal indepenence (i.e., $X^{(i)}(t), X_i(t')$ are independent between $t \neq t'$). While such an assumption is popular, allowing the temporal correlation widens the application of the framework. For example, the click probability of online advertisements depends on the period of time. There are some recent results that are potentially applicable in non-sparse high-dimensional regime (e.g., Nakakita and Imaizumi [2022]).

- **Exponentially decaying models:** AEtC directly assumes the eigenvalue structure of the covariance matrix (in particular, Example 1 or Example 2 in Proposition 1). Exploring the inclusion of exponentially decaying eigenvalues, such as Example 4 in Theorem 31 of Bartlett et al. [2020], would be an intriguing avenue to explore.

- **Nested models:** Foster et al. [2019] considered the contextual bandit problem under the nested model and introduced an algorithm that can adapt to the problem's complexity. Considering such a nested model should be interesting.

# B    Comparison with Sparse Bandits

Similar to our setting, sparse linear bandit models accept a very large number of features[10]. Sparse bandit algorithms employ the $\ell 1$ regularizer to suppress most coefficients. The regret of a sparse bandit algorithm is characterized by the number of non-zero features $s$. In our case, $s$ corresponds to $p$ since all dimensions exhibit non-zero values. Consequently, regret bounds in sparse bandit translate into $O(T)$ trivial bounds within our framework. We consider the sparse and benign overfitting to be orthogonal. There are numerous examples that are benign yet not sparse, and conversely, sparse but not benign. The primary incentive for benign overfitting theory is to illuminate the learnability of recent large-scale models. To rephrase, we are examining different problem classes in which we can ensure sublinear regret bounds, irrespective of any assumptions about sparsity.

# C    Proofs on Risk Bounds

We give proofs on the upper bound in Theorem 2. The bound is derived from the risk bound in Tsigler and Bartlett [2020] and adapted for our setting. A significant difference here is that the budget $T$, which characterizes the covariance matrices $\Sigma^{(i)}$, and the sample size $N$ used for estimation have different values.

We give some additional notation. For a vector $b \in \mathbb{R}^p$ and $q \in [p]$, let $b_{0:q} := (b_1, b_2, ..., b_q)$ is a sub-vector. Similarly, $b_{q:\infty} := (b_{q+1}, b_{q+2}, ..., b_p)$ is a sub-vector with the rest of the terms.

---

[10]Typically, the number of feature $p$ can be exponential to the number of datapoints $T$.

For a covariance matrix $\Sigma \in \mathbb{R}^{p \times p}$ with eigenvalues $\lambda_1, ..., \lambda_p$, $\Sigma_{0:q} = \text{diag}(\lambda_1, ..., \lambda_q)$ and $\Sigma_{q:\infty} = \text{diag}(\lambda_{q+1}, ..., \lambda_p)$ are diagonal matrices with the subset of eigenvalues. Similarly, for the data matrix $\mathbf{X}^{(i)} \in \mathbb{R}^{N \times p}$, $\mathbf{X}^{(i)}_{0:q}$ denotes a sub-matrix with the first $q$ columns of $\mathbf{X}^{(i)}$, and $\mathbf{X}^{(i)}_{q:\infty}$ denotes a sub-matrix with the rest of the columns $\mathbf{X}^{(i)}$.

*Proof of Theorem 2.* For each $i \in [K]$, we study the bound on the risk $\|\widehat{\theta}^{(i)} - \theta^{(i)}\|^2_{\Sigma^{(i)}}$. Using the fact that $\mathbf{Y}^{(i)} = \mathbf{X}^{(i)} \theta^{(i)} + \Xi^{(i)}$ with $\Xi^{(i)} = (\xi^{(i)}(1), ..., \xi^{(i)}(N))^\top \in \mathbb{R}^N$ where $\xi^{(i)}(t)$ is an i.i.d. copy of $\xi^{(i)}(1)$, we first decompose $\widehat{\theta}^{(i)}$ as

$$
\begin{aligned}
\widehat{\theta}^{(i)} &= (\mathbf{X}^{(i)})^\top (\mathbf{X}^{(i)}(\mathbf{X}^{(i)})^\top)^{-1} \mathbf{Y}^{(i)} \\
&= (\mathbf{X}^{(i)})^\top (\mathbf{X}^{(i)}(\mathbf{X}^{(i)})^\top)^{-1} \mathbf{X}^{(i)} \theta^{(i)} + (\mathbf{X}^{(i)})^\top (\mathbf{X}^{(i)}(\mathbf{X}^{(i)})^\top)^{-1} \Xi^{(i)} \\
&=: \widetilde{\theta}^{(i)} + \breve{\theta}^{(i)}.
\end{aligned}
$$

Using the decomposition, we decompose the total error as

$$
\|\widetilde{\theta}^{(i)} + \breve{\theta}^{(i)} - \theta^{(i)}\|^2_{\Sigma^{(i)}} \le 2\|\widetilde{\theta}^{(i)} - \theta^{(i)}\|^2_{\Sigma^{(i)}} + 2\|\breve{\theta}^{(i)}\|^2_{\Sigma^{(i)}} =: 2T_B + 2T_V.
$$

Here, $T_B$ denotes a bias term of the risk, and $T_V$ denotes a variance term.

In the following, we develop a bound on each of the terms. Fix $k \in [p]$. By Corollary 6 in Tsigler and Bartlett [2020], we achieve the following bounds with some constant $c > 0$, which holds with the desired probability. Here, we set $\mu_{n,k}(A_k^{-1})$ is the $n$-th largest eigenvalue of $A_k^{-1}$ for $n \in [N]$.

$$
T_B \le c\|\theta^{(i)}_{k:\infty}\|_{\Sigma^{(i)}_{k,\infty}} + c\|\theta^{(i)}_{0:k}\|_{(\Sigma^{(i)}_{0:k})^{-1}} \left( \frac{\sum_{j>k} \lambda^{(i)}_j}{N} \right)^2, \text{ and} \tag{13}
$$

$$
T_V \le c\frac{k}{N} + c\frac{N \sum_{j>k} (\lambda^{(i)}_j)^2}{(\sum_{j>k} \lambda^{(i)}_j)^2}. \tag{14}
$$

Note that only the number of samples $N$ appears explicitly in the boundary, while the budget $T$ affects it only implicitly through $\Sigma^{(i)}$.

We study the bound for the bias part in (13). Here, we set $k = k^*_N(\Sigma^{(i)}, N)$ using the coherent rank. By Hölder's inequality, we obtain

$$
\begin{aligned}
T_B &\le \|\theta^{(i)}_{k^*_N:\infty}\|^2_{\Sigma^{(i)}_{k^*_N:\infty}} + \|\theta^{(i)}_{0:k^*_N}\|^2_{\Sigma^{(i)-1}_{0:k^*_N}} \left( \frac{\sum_{j>k^*_N} \lambda^{(i)}_j}{N} \right)^2 \\
&\le \|\theta^{(i)}\|^2 \left( \lambda^{(i)}_{k^*_N+1} + (\lambda^{(i)}_{k^*_N})^{-1} \left( \frac{\sum_{j>k^*_N} \lambda^{(i)}_j}{N} \right)^2 \right) \\
&= \|\theta^{(i)}\|^2 \left( \lambda^{(i)}_{k^*_N+1} + (\lambda^{(i)}_{k^*_N})^{-1} \left( \frac{\sum_{j \ge k^*_N} \lambda^{(i)}_j - \lambda^{(i)}_{k^*_N}}{N} \right)^2 \right) \\
&= \|\theta^{(i)}\|^2 \left( \lambda^{(i)}_{k^*_N+1} + (\lambda^{(i)}_{k^*_N})^{-1} (\lambda^{(i)}_{k^*_N})^2 \left( \frac{\sum_{j \ge k^*_N} \lambda^{(i)}_j - \lambda^{(i)}_{k^*_N}}{\lambda^{(i)}_{k^*_N} N} \right)^2 \right) \\
&= \|\theta^{(i)}\|^2 \left( \lambda^{(i)}_{k^*_N+1} + \lambda^{(i)}_{k^*_N} \left( \frac{r_{k^*_N-1}(\Sigma^{(i)}) - 1}{N} \right)^2 \right),
\end{aligned}
$$

which follows the definition of the effective rank $r_k(\Sigma^{(i)})$. We also apply the properties of the ranks and obtain

$$
\|\theta^{(i)}\|^2 \left( \lambda^{(i)}_{k^*_N+1} + \lambda^{(i)}_{k^*_N} \left( \frac{r_{k^*_N-1}(\Sigma^{(i)}) - 1}{N} \right)^2 \right) \le \|\theta^{(i)}\|^2 \left( \lambda^{(i)}_{k^*_N+1} + \lambda^{(i)}_{k^*_N} \left( \frac{N-1}{N} \right)^2 \right)
$$

$$\leq \|\theta^{(i)}\|^2 \left( \lambda^{(i)}_{k^*_N+1} + \lambda^{(i)}_{k^*_N} \right)$$

$$\leq c' \|\theta^{(i)}\|^2 \lambda^{(i)}_{k^*_N}$$

$$\leq c' \|\theta^{(i)}\|^2 B^{(i)}_{N,T},$$

where $c' := (1 + b^2)$ is a constant. The second last inequality follows $\lambda^{(i)}_{k^*_N+1} \leq \lambda^{(i)}_{k^*_N}$. About the variance term $T_V$ in (14), we simply obtain $T_V \leq V^{(i)}_{N,T}$ by their definitions.

Finally, we set $C_U = \max\{c'\|\theta^{(i)}\|^2, c\}$ and obtain the statement. $\qquad\square$

## D  Proofs for EtC

### D.1  Concentration of the Maximum Value

**Lemma 10.** *Let $Y_1, Y_2, \ldots, Y_K$ be $K$ sub-Gaussian random variables with common parameter $\sigma$. Let $Y = \max_{i \in [K]} Y_i$ be the maximum of them [11]. Then, we have*

$$\mathbb{E}[Y] \leq \sigma \sqrt{2 \log K}. \tag{15}$$

*Proof of Lemma 10.* For any $\lambda > 0$, we have

$$e^{\lambda \mathbb{E}[\max_{i \in [K]} Y_i]} \leq \mathbb{E}\left[ e^{\lambda \max_{i \in [K]} Y_i} \right] \quad \text{(by Jensen's inequality)} \tag{16}$$

$$= \mathbb{E}\left[ \max_{i \in [K]} e^{\lambda Y_i} \right] \tag{17}$$

$$\leq \sum_i \mathbb{E}[e^{\lambda Y_i}] \leq N e^{\lambda^2 \sigma^2/2}, \tag{18}$$

and thus

$$\mathbb{E}[\max_{i \in [K]} Y_i] \leq \frac{\log N}{\lambda} + \frac{\lambda \sigma^2}{2},$$

and taking $\lambda = \sqrt{2 \log N / \sigma^2}$ yields the result. $\qquad\square$

### D.2  High-probability Confidence Bound

**Lemma 11.** *There exists $C > 0$ such that, for any $T_0 \geq CK \log T$, with probability at least $1 - O(T^{-1})$, event*

$$\mathcal{A} = \bigcap_{i \in [K]} \mathcal{A}_i \tag{19}$$

$$\mathcal{A}_i = \left\{ \|\widehat{\theta}^{(i)} - \theta^{(i)}\|_{\Sigma^{(i)}} \leq C \, \mathrm{E}(T_0/K, T) \right\}. \tag{20}$$

*holds. Moreover, under $\mathcal{A}$, we have*

$$\mathbb{E}[\langle X^{(i^*(t))}(t), \theta^{(i^*(t))} \rangle - \langle X^{(I(t))}(t), \theta^{(I(t))} \rangle] \leq 2C \, \mathrm{E}(T_0/K, T) \sqrt{2 \log K} \tag{21}$$

*for each $t = T_0 + 1, T_0 + 2, \ldots, T$.*

Note that, in EtC, we have uniform exploration over $K$ arms, and thus $N = T_0/K$.

*Proof of Lemma 11.* Theorem 2 implies that for some $C > 0$, with probability at least $1 - c_U e^{-(C \log T)/c_U} \geq 1 - 1/T$, $\mathcal{A}_i$ holds. Moreover, the union bound of this over $K$ arms implies $\mathcal{A}$ holds with probability $1 - K/T$.

Using the definition of the optimal arm $i^*(t) = \mathrm{argmax}_{i^*}\langle X_{i^*}(t), \theta_{i^*} \rangle$, we have

$$\langle X^{(i^*(t))}(t), \theta^{(i^*(t))} \rangle - \langle X^{(I(t))}(t), \theta^{(I(t))} \rangle$$

---

[11]These random variables can be dependent each other.

$$\leq \langle X^{(i^*(t))}(t), \widehat{\theta}_{i^*(t)} \rangle - \langle X^{(I(t))}(t), \widehat{\theta}_{I(t)} \rangle + 2D_{I(t)}(t)$$

$$\leq 2D_{I(t)}(t) \quad \text{(by definition of } I(t))$$

$$= 2 \max_{i \in [K]} D_i(t),$$

where $\Delta_i = \theta^{(i)} - \widehat{\theta}^{(i)}$ and $D_i(t) = \langle X_i(t), \Delta_i \rangle$. Given $\Delta_i$ at the end of round $T_0$, $D_i(t)$ for each $i$ is a sub-Gaussian random variable. Under the event $\mathcal{A}$, the variance of $D_i(t)$ is bounded as

$$\mathbb{E}[|D_i(t)|] \leq C \, \mathrm{E}(T_0/K, T). \tag{22}$$

By using this and Lemma 10 on the maximum of $K$ sub-Gaussian random variables, we have,

$$2\mathbb{E}[|D(t)|] = 2\mathbb{E}[|\max_i D_i(t)|] \tag{23}$$

$$\leq 2C \, \mathrm{E}(T_0/K, T)\sqrt{2\log K}. \quad \text{(by Lemma 10)} \tag{24}$$

$\square$

### D.3 Proof of Theorem 4

By using Lemma 11, we derive the regret upper bound of Theorem 4.

*Proof of Theorem 4.* The regret (Eq. (2)) is bounded as

$$R(T) = \mathbb{E}\left[\sum_{t=1}^{T} \langle X^{(i^*(t))}(t), \theta^{(i^*(t))} \rangle - \langle X^{(I(t))}(t), \theta^{(I(t))} \rangle\right]$$

$$= \sum_{t=1}^{T_0} \mathbb{E}[\langle X^{(i^*(t))}(t), \theta^{(i^*(t))} \rangle - \langle X^{(I(t))}(t), \theta^{(I(t))} \rangle]$$

$$+ \sum_{t=T_0+1}^{T} \mathbb{E}[\langle X^{(i^*(t))}(t), \theta^{(i^*(t))} \rangle - \langle X^{(I(t))}(t), \theta^{(I(t))} \rangle],$$

$$=: R_1 + R_2.$$

We bound the first term $R_1$ as

$$R_1 \leq \sum_{t=1}^{T_0} 2\mathbb{E}\left[\max_{i \in [K]} \langle X^{(i)}(t), \theta^{(i)} \rangle\right]$$

$$\leq \sum_{t=1}^{T_0} 2\kappa_x \sqrt{2\log K}\theta_{\max}$$

$$\text{(by Assumption 1 and Lemma 10)}$$

$$= T_0 \times 2\kappa_x\theta_{\max}\sqrt{2\log K} = \widetilde{O}(T_0).$$

We bound the second term $R_2$ as

$$R_2 \leq \sum_{t=T_0+1}^{T} \mathbb{E}\left[\mathbf{1}\{\mathcal{A}\}2C \, \mathrm{E}(T_0/K, T)\sqrt{2\log K} + \mathbf{1}\{\mathcal{A}^c\}(\langle X^{(i^*(t))}(t), \theta^{(i^*(t))} \rangle - \langle X^{(I(t))}(t), \theta^{(I(t))} \rangle)\right]$$

$$\text{(by Eq. (21))}$$

$$\leq \sum_{t=T_0+1}^{T} \mathbb{P}(\mathcal{A})2C \, \mathrm{E}(T_0/K, T)\sqrt{2\log K} + \mathbb{P}(\mathcal{A}^c) \times 2\kappa_x\theta_{\max}\sqrt{2\log K}$$

$$\text{(by the same discussion as } R_1)$$

$$\leq 2TC \, \mathrm{E}(T_0/K, T)\sqrt{2\log K} + \widetilde{O}(T)\mathbb{P}(\mathcal{A}^c)$$

$$\text{(by Lemma 11)}$$

$$\le 2TC\,\mathrm{E}(T_0/K,T)\sqrt{2\log K} + \widetilde{O}(T) \times \frac{K}{T}.$$

Combining these bounds, we obtain

$$R(T) = R_1 + R_2 \le \widetilde{O}(T_0) + \widetilde{O}(T\,\mathrm{E}(T_0/K,T)).$$

This bound is optimized by choosing $T_0$ in accordance with Eq. (7), and then we have the theorem. $\quad\square$

### D.4 Proof of Theorem 5

This section provides the regret lower bound of EtC.

*Proof of Theorem 5.* We consider a model with $K = 3$. Let $\epsilon_T = \mathrm{E}(T_0/K,T)$. We explicitly construct $\Sigma^{(i)}$ as follows: First, $\Sigma^{(1)}, \Sigma^{(2)}, \Sigma^{(3)}$ are identical and benign, and denote $k$-th eigenvalue as $\lambda_k^{(i)} = \lambda_k$. The coefficient $\theta^{(1)} = (1,0,0,\dots)^\top$, $\theta^{(2)} = (\epsilon_T,0,0,\dots)^\top$, and $\theta^{(3)} = (0,0,0,\dots)^\top$. With these $\{\theta^{(i)}\}_{i=1,2,3}$, the only non-zero coefficients are $\theta_1^{(1)}, \theta_1^{(2)}$, and thus the best arm is defined in terms of $X_1^{(1)}(t), X_1^{(2)}(t)$, which are the first components of $X^{(1)}(t), X^{(2)}(t)$, respectively.

Let $R_1, R_2$ be regret during the exploration and exploitation phases, respectively.

**Regret during the exploration phase:**
We first bound the regret during the exploration phase where we draw an arm uniformly randomly. In the following, we show that the regret per round is $\Omega(1)$. Since $X^{(i)}(t)$ is a zero-mean ($p$-dimensinal) Gaussian, its linear function such as $\langle X^{(i)}(t), \theta^{(i)}\rangle$ are $\langle X^{(i)}(t), \widehat{\theta}^{(i)}\rangle$ are zero-mean univariate Gaussians given $\theta^{(i)}, \widehat{\theta}^{(i)}$. Therefore,

$$r_{ij}(t) := \langle X^{(i)}(t), \theta^{(i)}\rangle - \langle X^{(j)}(t), \theta^{(j)}\rangle$$

is also a Gaussian, and its variance is $\Theta(1)$. Therefore, $\mathbb{P}[r_{12}(t) \ge 1], \mathbb{P}[r_{13}(t) \ge 1] = \Theta(1)$. The regret in the rounds we draw arm $2,3$ is at least $\mathbf{1}[r_{12}(t) \ge 1]r_{12}(t), \mathbf{1}[r_{13}(t) \ge 1]r_{13}(t) = \Theta(1)$, which implies the regret-per-round is $\Omega(1)$.

In summary, $\mathbb{E}[R_1] = \Omega(1) \times T_0 = \Omega(T_0)$.

**Regret during the exploitation phase:**
We then bound the regret during the exploitation phase. Intuitively speaking, an algorithm misidentifies the better arm among 2 and 3 with $\Omega(1)$ probability and the regret per such a misidentification is $\Omega(\epsilon_T)$.

Theorems 2 and Lemma 3 state that there exists a constant $c > 0$ such that probability at least $1 - ce^{-N/c}$, for all $i$ we have

$$\mathbb{E}\left[\|\widehat{\theta}^{(i)} - \theta^{(i)}\|_{\Sigma^{(i)}}\right] \in (C^L\epsilon_T, C^U\epsilon_T) \tag{25}$$

for some constants $C^L, C^U > 0$.

Without loss of generality, we assume $N$ to be sufficiently large such that $1 - ce^{-N/c} > 1/2$ because if $N = O(\log T)$ then the size of the error bound is at least polylogarithmic, which results in regret of $\widetilde{\Omega}(T)$. In the following, we assume Eq. (25) is the case, and we show that the regret-per-round is $\Omega(\epsilon_T)$. We define the events as follows:

$$\mathcal{G}(t) := \{X_1^{(2)}(t) \in [1,2]\} \tag{26}$$

$$\mathcal{H}(t) := \{\langle X_1^{(1)}(t), \widehat{\theta}^{(1)}\rangle < \langle X_1^{(2)}(t), \widehat{\theta}^{(2)}\rangle < \langle X_1^{(3)}(t), \widehat{\theta}^{(3)}\rangle\}. \tag{27}$$

Event $\mathcal{G}(t)$ states that arm 2 is positive and not very large, Event $\mathcal{H}(t)$ states that the algorithm draws considers the arm 3 is the best and arm 2 is the second best.

If $\mathcal{G}(t) \cap \mathcal{H}(t)$ is the case, then arm 3 is chosen but suboptimal, and the regret is at least

$$\langle X_1^{(2)}(t), \theta_1^{(2)}\rangle - \langle X_1^{(3)}(t), \theta_1^{(3)}\rangle \in [\epsilon_T, 2\epsilon_T], \tag{28}$$

where we used $\mathcal{G}(t)$, $\langle X^{(2)}(t), \theta^{(2)}\rangle = X_1^{(2)}(t)\epsilon_T$, and $\langle X^{(3)}(t), \theta^{(3)}\rangle = 0$. Therefore, showing

$$\mathbb{P}[\mathcal{G}(t) \cap \mathcal{H}(t)] = \Omega(1) \tag{29}$$

suffices to show that regret-per-round is $\Omega(\epsilon_T)$. First, $X_{2,1}(t)$ is a Gaussian with scale $\Theta(1)$, which implies $\mathcal{G}$ occurs with probability $\Theta(1)$. Second, conditioned on $\mathcal{G}(t)$, the sufficient condition for $\mathcal{H}(t)$ is

$$\mathcal{H}'(t) := \{\langle X^{(1)}(t), \widehat{\theta}^{(1)}\rangle < 0, \langle X^{(3)}(t), \widehat{\theta}^{(3)}\rangle > 0, \langle X^{(3)}(t), \widehat{\theta}^{(3)}\rangle - \langle X_{2,\backslash 1}^{(2)}(t), \widehat{\theta}_{\backslash 1}^{(2)}\rangle \geq 2\epsilon_T\}, \quad (30)$$

where

$$\langle X_{2,\backslash 1}^{(2)}(t), \widehat{\theta}_{\backslash 1}^{(2)}\rangle := \langle X^{(2)}(t), \widehat{\theta}^{(2)}\rangle - X_1^{(2)}(t)\,\widehat{\theta}_1^{(2)},$$

is an inner product that ignores the first component. Here, under Eq. (25), each of $i \in [3]$, $\langle X^{(i)}(t), \widehat{\theta}^{(i)}\rangle$, and $\langle X_{2,\backslash 1}^{(2)}(t), \widehat{\theta}_{\backslash 1}^{(2)}\rangle$, is a Gaussian random variables with its standard deviation $\Omega(\epsilon_T)$. By this fact it is clear that $\mathbb{P}[\mathcal{H}'(t)] = \Omega(1)$. Therefore, $\mathbb{P}[\mathcal{H}'(t)|\mathcal{G}(t)] = \Omega(1)$. In summary, $\mathcal{G}(t) \cap \mathcal{H}(t) \supset \mathcal{G}(t) \cap \mathcal{H}'(t)$ occurs with probability $\Omega(1)$, and thus regret-per-round is $\Omega(\epsilon_T)$ by Eq. (28).

In summary, $\mathbb{E}[R_2] \geq \Omega(\epsilon_T) \times T$, and the expected regret is bounded as

$$R(T) = \mathbb{E}[R_1] + \mathbb{E}[R_2] \tag{31}$$
$$= \Omega(T_0) + \Omega(\epsilon_T)T, \tag{32}$$

which completes the proof. $\qquad\square$

# E   Proofs for AEtC

## E.1   Proof of Lemma 6

We have

$$|\mathrm{tr}(\Sigma^{(i)}) - \mathrm{tr}(\widehat{\Sigma}^{(i)})| \tag{33}$$

$$= \log(T) \times O\left(\sqrt{\sum_j \lambda_j^2}\right) \tag{34}$$

(by Lemma 12 with $\eta = O(\log(T))$, transformation above follows with probability $\mathrm{poly}(T^{-1})$) (35)

$$= \begin{cases} \widetilde{O}(1) & (\text{Example 1, where we used } \sum_j (j^{-(1+)})^2 = O(1)), \\ \widetilde{O}\left(T^{c(1-2b)/2}\right) & (\text{Example 2, where we used } \sum_{j=1}^{T^c}(j^{-b})^2 = [T^c]^{1-2b}), \end{cases} \tag{36}$$

$$= o\left(\mathrm{tr}(\Sigma^{(i)})\right),$$

(by $\mathrm{tr}(\Sigma^{(i)}) = O(T^a)$ (Example 1) or $\mathrm{tr}(\Sigma^{(i)}) = O(T^{c(1-b)})$ (Example 2)) (37)

from which Lemma 6 follows.

**Lemma 12.** *Suppose that Assumption 1 holds. Assume that $\lambda_1, \sum_j \lambda_j < +\infty$. Let the empirical covariance matrix with $N$ samples be*

$$\widehat{\Sigma}^{(i)} := \frac{1}{N}(\mathbf{X}^{(i)})^\top \mathbf{X}^{(i)}. \tag{38}$$

*Then, for any $\eta > 1$, with probability at least $1 - 2e^{-\eta}$, we have*

$$|\mathrm{tr}(\widehat{\Sigma}^{(i)}) - \mathrm{tr}(\Sigma^{(i)})| \leq \frac{C\kappa_x^2 \max\left(\eta\lambda_1, \sqrt{\eta\sum_j \lambda_j^2}\right)}{N} \leq C\kappa_x^2 \eta\sqrt{\sum_j \lambda_j^2}$$

*for some constant $C > 0$.*

*Proof of Lemma 12.* We have

$$\mathrm{tr}(\widehat{\Sigma}^{(i)}) = \frac{1}{N}\mathrm{tr}\left((\mathbf{X}^{(i)})^\top \mathbf{X}^{(i)}\right) \tag{39}$$

$$= \frac{1}{N} \text{tr} \left( V^{(i)} (\Lambda^{(i)})^{1/2} (\mathbf{Z}^{(i)})^\top \mathbf{Z}^{(i)} (\Lambda^{(i)})^{1/2} (V^{(i)})^\top \right) \tag{40}$$

$$= \frac{1}{N} \text{tr} \left( (\Lambda^{(i)})^{1/2} (V^{(i)})^\top V^{(i)} (\Lambda^{(i)})^{1/2} (\mathbf{Z}^{(i)})^\top \mathbf{Z}^{(i)} \right) \tag{41}$$

$$= \frac{1}{N} \text{tr} \left( \Lambda^{(i)} (\mathbf{Z}^{(i)})^\top \mathbf{Z}^{(i)} \right), \tag{42}$$

and thus

$$\text{tr}(\widehat{\Sigma}^{(i)}) - \text{tr}(\Sigma^{(i)}) = \frac{1}{N} \text{tr} \left( \Lambda^{(i)} \left( (\mathbf{Z}^{(i)})^\top \mathbf{Z}^{(i)} - I_p \right) \right). \tag{43}$$

Each diagonal element of $(\mathbf{Z}^{(i)})^\top \mathbf{Z}^{(i)} - I_p$ is a sum of $N$ independent samples, and each sample is zero-mean $\kappa_x^2$ sub-exponential random variable. Using Lemma 12 in Bartlett et al. [2020], with probability at least $1 - 2e^{-\eta}$, we have

$$\left| \text{tr} \left( \Lambda^{(i)} \left( (\mathbf{Z}^{(i)})^\top \mathbf{Z}^{(i)} - I_p \right) \right) \right| \le C \kappa_x^2 \max \left( \eta \lambda_1, \sqrt{\eta \sum_j \lambda_j^2} \right), \tag{44}$$

for some $C > 0$, which completes the proof. □

## E.2  Proof of Lemma 7

**Lemma 13.** (Lemma 4.2 in Bosq [2000]) *For any two matrices* $\mathbf{X}_0, \mathbf{X}_1$ *with their eigenvalues* $(\lambda_{0,j})_{j \in [p]}$ *and* $(\lambda_{1,j})_{j \in [p]}$*, we have*

$$|\lambda_{0,j} - \lambda_{1,j}| \le \|\mathbf{X}_0 - \mathbf{X}_1\|_{\text{op}} \ \forall_{j \in [p]}. \tag{45}$$

**Lemma 14** (Corollary 2 in Koltchinskii and Lounici [2017]). *Suppose that* $X_1, ..., X_N$ *are* $\mathbb{R}^p$*-valued i.i.d. random vectors whose mean is zero and covariance is* $\Sigma$. $\widehat{\Sigma} = N^{-1} \sum_{i=1}^N X_i X_i^\top$ *is an empirical covariance matrix. Then, there exists a constant* $C > 0$ *and with probability* $1 - e^{-\eta}$ *we have*

$$\|\widehat{\Sigma} - \Sigma\|_{\text{op}} \le C \|\Sigma\|_{\text{op}} \sqrt{\frac{\widetilde{\mathbf{r}}(\Sigma) + \eta}{N}}, \tag{46}$$

*where*

$$\widetilde{\mathbf{r}}(\Sigma) := \frac{\text{tr}(\Sigma)}{\|\Sigma\|_{\text{op}}}.$$

*Proof of Lemma 7.* Lemmas 13 and 14 imply that

$$\max_j |\widehat{\lambda}_j - \lambda_j| \le \|\widehat{\Sigma} - \Sigma\|_{\text{op}} = C \sqrt{\frac{\text{tr}(\Sigma^{(i)}) + \eta}{N}},$$

with probability $1 - e^{-\eta}$, and setting $\eta = C_{\text{poly}} \log T$ yields the desired result. □

## E.3  Proof of Lemma 8

This section adopts the same set of assumptions as Lemma 8.

**Lemma 15.** *Let* $\tau$ *be a constant that is independent of* $T$. *Then, for any* $C_{\text{poly}} > 0$*, with probability at least* $1 - T^{C_{\text{poly}}}$*, we have*

$$|\beta_T - \widehat{\beta}_T| = O \left( \sqrt{\frac{\text{tr}(\Sigma^{(i)}) + C_{\text{poly}} \log T}{N}} \right). \tag{47}$$

*Proof of Lemma 15.* We assume Eq. (10) that holds with probability $1 - T^{C_{\text{poly}}}$. We have,

$$\beta_T - \widehat{\beta}_T := \frac{1}{\tau} \sum_{k=1}^\tau \frac{\left( \log(\lambda_k / \lambda_{k+1}) - \log(\widehat{\lambda}_k / \widehat{\lambda}_{k+1}) \right)}{\log((k+1)/k)} \tag{48}$$

$$= \frac{1}{\tau} \sum_{k=1}^{\tau} \frac{\left( \beta_T \log(\frac{k+1}{k}) - \log\left( \frac{(k+1)^{\beta_T}}{k^{\beta_T}} + C\sqrt{\frac{\text{tr}(\Sigma^{(i)}) + C_{\text{poly}} \log T}{N}} \right) \right)}{\log((k+1)/k)} \tag{49}$$

$$= O\left( \sqrt{\frac{\text{tr}(\Sigma^{(i)}) + C_{\text{poly}} \log T}{N}} \right), \tag{50}$$

where we have used $\log(c+x) = \log(c) + \log(1+x/c) \approx \log(c) + \frac{x}{c} + o(x)$ in the last transformation. $\quad\square$

**Lemma 16.** *For $1 \leq k \leq N^{(i)}$ and any $C_{\text{poly}} > 0$, with probability at least $1 - T^{-C_{\text{poly}}}$, we have*

$$\log\left( \frac{\widehat{r}_k(\Sigma^{(i)})}{r_k(\Sigma^{(i)})} \right) = \log(T^{o(1)}).$$

*Moreover, we have*

$$\log\left( \frac{\widehat{k}_n}{k_n^*} \right) = \log(T^{o(1)})$$

*for any $i \in [K], n \in [N^{(i)}]$.*

*Proof of Lemma 16.* We first show the first $k$ eigenvalues are negligible for $k = O(T)$: Namely,

$$\frac{\sum_{j<k} \lambda_j^{(i)}}{\text{tr}(\Sigma^{(i)})} = \begin{cases} O\left( \frac{T^a(k^{1/T^a}-1)}{T^a} \right) & \text{(Example 1),} \\ O\left( \frac{k^{1-b}}{T^{c(1-b)}} \right) & \text{(Example 2),} \end{cases} = o(1). \tag{51}$$

Moreover, we have

$$\log\left( \frac{\widehat{r}_k(\Sigma^{(i)})}{r_k(\Sigma^{(i)})} \right) = \log\left( \left( \frac{\text{tr}(\widehat{\Sigma}^{(i)})}{\sum_{j>k} \lambda_j^{(i)}} \right) \cdot \left( \frac{\lambda_{k+1}^{(i)}}{\widehat{\lambda}_{k+1}^{(i)}} \right) \right) \tag{52}$$

$$= \log\left( \left( \frac{\text{tr}(\widehat{\Sigma}^{(i)})}{\text{tr}(\Sigma^{(i)})} \right) \cdot \left( \frac{\lambda_{k+1}^{(i)}}{\widehat{\lambda}_{k+1}^{(i)}} \right) \right) + o(1) \quad \text{(by Eq. (51))} \tag{53}$$

$$= \log\left( \left( \frac{\text{tr}(\widehat{\Sigma}^{(i)})}{\text{tr}(\Sigma^{(i)})} \right) \cdot \left( \frac{(k+1)^{-\beta_T}}{(k+1)^{-\widehat{\beta}_T}} \right) \right) + o(1) \tag{54}$$

$$= \log\left( \left( \frac{\text{tr}(\Sigma^{(i)})}{\text{tr}(\Sigma^{(i)})} \right) \cdot \left( \frac{(k+1)^{-\beta_T}}{(k+1)^{-\widehat{\beta}_T}} \right) \right) + o(1) \quad \text{(by Lemma 6)} \tag{55}$$

$$= \log\left( \left( \frac{\text{tr}(\Sigma^{(i)})}{\text{tr}(\Sigma^{(i)})} \right) \cdot \left( \frac{(k+1)^{-\beta_T}}{(k+1)^{-\beta_T}} \right) \right) + \log(T^{o(1)}) + o(1) \quad \text{(by } k+1 = O(T) \text{ and Lemma 15)} \tag{56}$$

$$= \log(T^{o(1)}). \tag{57}$$

By definition,

$$k^* = \min\{k \geq 0 \mid r_k(\Sigma^{(i)}) \geq N\} \tag{58}$$

$$\widehat{k}_N = \min\{k \geq 0 \mid \widehat{r}_k(\Sigma^{(i)}) \geq N\}. \tag{59}$$

Eq. (57) states that $r_k(\Sigma^{(i)})/\widehat{r}_k(\Sigma^{(i)}) = T^{o(1)}$, and by using the fact that $\lambda_k^{(i)} = k^{-a}$ (Example 1) or $\lambda_k^{(i)} = k^{-b}$ (Example 2), we have $k^*/\widehat{k}_N = T^{o(1)}$, which is equivalent to

$$\log\left( \frac{k^*}{\widehat{k}_N} \right) = \log(T^{o(1)}). \tag{60}$$

$\square$

**Lemma 17.** *For any $n \in [N]$, with probability at least $1 - T^{-C_{\text{poly}}}$, we have*

$$\log\left(\frac{\widehat{R}_{\widehat{k}_n}(\Sigma^{(i)})}{R_{k_n^*}(\Sigma^{(i)})}\right) = \log(T^{o(1)}).$$

*Proof of Lemma 17.* As same to the proof of Lemma 16, we use Lemma 7 and utilize $|\widehat{\lambda}_j^{(i)} - \lambda_j^{(i)}| \leq C\sqrt{\frac{\text{tr}(\Sigma^{(i)}) + C_{\text{poly}} \log T}{N}}$ for every $j = 1, ..., p$, which holds with probability $1 - T^{-C_{\text{poly}}}$. Eq. (60) implies that

$$\log\left(\frac{\widehat{R}_{\widehat{k}_n}(\Sigma^{(i)})}{\widehat{R}_{k_n^*}(\Sigma^{(i)})}\right) = T^{o(1)}.$$

Then, we study the ratio as

$$\log\left(\frac{\widehat{R}_{k_n^*}(\Sigma^{(i)})}{R_{k_n^*}(\Sigma^{(i)})}\right) = \log\left(\left(\frac{\sum_{j>k_n^*} \lambda_j^{(i)}}{\text{tr}(\widehat{\Sigma}^{(i)})}\right)^2 \frac{\sum_{j>\widehat{k}_n} (\widetilde{\lambda}_j^{(i)})^2}{\sum_{j>k_n^*} (\lambda_j^{(i)})^2}\right)$$

$$= \log\left(\left(\frac{\text{tr}(\Sigma^{(i)})}{\text{tr}(\widehat{\Sigma}^{(i)})}\right)^2 \frac{\sum_{j>\widehat{k}_n} (\widetilde{\lambda}_j^{(i)})^2}{\sum_{j>k_n^*} (\lambda_j^{(i)})^2}\right) + o(1) \quad \text{(by Eq. (51))}$$

$$= \log\left(\left(\frac{\text{tr}(\Sigma^{(i)})}{\text{tr}(\Sigma^{(i)})}\right)^2 \frac{\sum_{j>\widehat{k}_n} (\widetilde{\lambda}_j^{(i)})^2}{\sum_{j>k_n^*} (\lambda_j^{(i)})^2}\right) + o(1) \quad \text{(by Lemma 6)}$$

$$= \log\left(\left(\frac{\text{tr}(\Sigma^{(i)})}{\text{tr}(\Sigma^{(i)})}\right)^2 \frac{\sum_{j>\widehat{k}_n} (\lambda_j^{(i)})^2}{\sum_{j>k_n^*} (\lambda_j^{(i)})^2}\right) + \log(T^{o(1)}) + o(1) \quad \text{(by Lemma 15)}$$

$$= \log\left(\left(\frac{\text{tr}(\Sigma^{(i)})}{\text{tr}(\Sigma^{(i)})}\right)^2 \frac{\sum_{j>k_n^*} (\lambda_j^{(i)})^2}{\sum_{j>k_n^*} (\lambda_j^{(i)})^2}\right) + \log(T^{o(1)}) \quad \text{(by Eq. (60))}$$

$$= \log(T^{o(1)}).$$

$\square$

*Proof of Lemma 8.* The statement is straightforward from Lemmas 16 and 17. $\square$