# OpenReview forum: "High-dimensional Contextual Bandit Problem without Sparsity"
_NeurIPS.cc/2023/Conference — NeurIPS 2023 poster_

### Official Review · Reviewer_57ot · 2023-06-10

**Soundness:** 2 fair
**Presentation:** 2 fair
**Contribution:** 2 fair
**Rating:** 5
**Confidence:** 3

**Summary:**

The paper addresses the problem of high-dimensional linear contextual bandits where the number of features is greater than the budget or even infinite. Unlike previous works, the paper does not impose sparsity on the regression coefficients but instead leverages over parameterized models. The authors propose an explore-then-commit (EtC) algorithm and an adaptive explore-then-commit (AEtC) algorithm to tackle this problem. The performance of the proposed algorithms is evaluated through simulations, and the optimal rate of the ETC algorithm is derived, showing that it can be achieved by balancing exploration and exploitation.

**Strengths:**

The paper studies a new problem in high-dimensional bandits when there is no sparsity in parameter vector. EtC is first proposed based on the minimum-norm interpolating estimator with the optimal exploration rate, and then AEtC is introduced to deal with the model-dependent parameters on the covariance.

**Weaknesses:**

1. Lower bound of the problem is not discussed in the paper. Since this is the first work on the high-dimensional bandit without sparsity, a lower bound can help people understand how good your algorithm is in theory. I acknowledge that authors mention this point as limitation in the appendix, but I believe this is an important issue.

2. Assumptions are strong and more explanations will be better. Some assumptions are given in the paper, e.g. Assumption 1, which may not be realistic in practice. For example, I can hardly understand the first part of Assumption 1. What is the intuition behind it? Why you used spectral decomposition to construct $X^{(i)}(t)$? Some examples on which these assumptions hold should be given.

3. More explanation on definitions should be presented. Authors use some definitions in this work that are not commonly used in the bandit community, e.g. effective bias, effective variance, coherent rank, effective rank, benign covariance. Some more explanations on these definitions and maybe correlation between them should be given in the paper.

4. The regret bound in Theorem 4 depends on some $\alpha > 0$, but this value is not clearly explained in Theorem 4. I think when authors write a formal Theorem in the paper, it is better to explain every term in detail (e.g. how to formulate $\alpha$ in general).

5. (minor): in the first page, the footnote 1 is under the sentence "submitted to 37th Conference on .......", which seems unusual. Maybe the authors change the margin of the template a little bit?

**Questions:**

Besides the questions in the above Weakness section, I have some questions here:

1. For your experiments, in several cases (DGP 1, DGP 2) it seems that the cumulative regret of your algorithm and other existing algorithms is linearly increasing according to the plots. Could authors explain why sublinear pattern is not observed?

2. Can your algorithm still work well when $p$ is greater than $T$ in practice? It might be more comprehensive if such case could be considered in your simulation studies.

**Limitations:**

No negative societal impact.

---

> ### Author Rebuttal · Authors · 2023-08-10
>
> We thank your careful reading and insightful comments. The score is a bit low even though no critical issue was raised. If the main concerns are the unclarity of the problem setting, we would be happy to address any concerns. Regarding the clarification on the benign covariance, we will revise the paper in accordance with the global reply above. Below, we answer each of the questions.
>
> > Lower bound of the problem is not discussed in the paper.
>
> Thank you for your clarification. We derived the lower bound on the class of EtC algorithms that matches our regret upper bound. You are right that this does not exclude the possibility of non-EtC algorithms that has a lower regret. We would also like to mention that the sparse linear bandit problem was introduced without a nontrivial lower bound (Abbasi-Yadkori et al. 2012). Given that we introduced a new problem, non-adaptive (EtC) and adaptive (AEtC) algorithms as well as experimental results, we consider the contributions of this paper are more than what is expected.
>
> Yasin Abbasi-Yadkori, David Pal, Csaba Szepesvari. “Online-to-Confidence-Set Conversions and Application to Sparse Stochastic Bandits.” (Sparse linear bandits)
>
> > Assumptions are strong, and more explanations will be better. Some assumptions are given in the paper, e.g. Assumption 1, which may not be realistic in practice. For example, I can hardly understand the first part of Assumption 1. What is the intuition behind it?
>
> The combination of Assumption 1 and Definition 1 is from the theory of benign overfitting (Bartlett et al 2020, Tsigler and Bartlett, 2020). We clarify that the first assumption implies that each feature is an independent sub-Gaussian after diagonalization. Covariance matrices in Examples 1 and 2 satisfy these assumptions. The second assumption implies that the rewards are noisy with subGaussian noise. Note that sub-Gaussian noises are standard in this assumption. Thank you for your clarification questions. We will revise the paper accordingly.
>
> > More explanation on definitions should be presented. Authors use some definitions in this work that are not commonly used in the bandit community, e.g. effective bias, effective variance, coherent rank, effective rank, benign covariance. Some more explanations on these definitions and maybe correlation between them should be given in the paper.
>
> Thank you for your suggestions. Since we introduce the new problem into the bandit community, we will add a more detailed discussion on these terminologies as outline in the global reply above. Effective rank characterizes the complexity of the problem where the spectrum is taken into consideration. Eigenvalues that are close to zero do little contribution to the effective rank. Coherent rank splits eigenvalues into major and minor ones. Bias corresponds to loss due to the minor eigenvalues that are unlearnable but acceptable. Variance corresponds to the loss from the major eigenvalues that are learnable.
>
> > The regret bound in Theorem 4 depends on some, but this value is not clearly explained in Theorem 4
>
> $T^{\alpha}$ is such that $N = T E(N, T)$ (Eq.(7) with $K=O(1)$), and its closed formula, which depends on the covariance spectrum, is derived in Examples 1 and 2. The value $\alpha$ in Thm 4 and 9 is identical. We will revise the paper to clarify this.
>
> > For your experiments, in several cases (DGP 1, DGP 2) it seems that the cumulative regret of your algorithm and other existing algorithms is linearly increasing according to the plots. Could authors explain why sublinear pattern is not observed?
>
> EtC has two phases (exploration and commitment), and it has a linear growth of regret for each of the phases for a fixed $T$. Sublinear regret (Theorem 9) implies that, if we run the EtC for each of $T=1,2,4,8,\dots$, $R(T)$ (regret at the final round) grows sublinear to $T$.
>
> > Can your algorithm still work well when is greater than in practice?
>
> We can deal with $p > T$ or even infinite $p= \infty$ as long as our assumptions hold. Example 1 is such a case.
>
> We appreciate many other suggestions.

---

> > ### Comment · Reviewer_57ot · 2023-08-10
> > **Thanks for your response**
> >
> > Now I have a much better understanding of different elements and definitions in your paper (I feel your original paper is technically too fast for most readers in bandit community). I will first raise my rating to 4 now. These days I will spend some more time reading your responses to other reviews' concerns and then adjust my rating accordingly. Thanks.

---

> > > ### Author Response · Authors · 2023-08-11
> > > **Thank you for the update**
> > >
> > > We sincerely value the time you've dedicated to reviewing our work. We acknowledge that some parts of our paper might require more explanation, particularly due to our use of tools that may not be commonly known within the bandit community. We are more than happy to provide further details or clarification to address your concerns.

---

> > > > ### Comment · Reviewer_57ot · 2023-08-18
> > > >
> > > > Thank you for your detailed explanation. I have no question now, and I just raised my rating to acceptance.

---

### Official Review · Reviewer_C3HD · 2023-07-05

**Soundness:** 3 good
**Presentation:** 3 good
**Contribution:** 2 fair
**Rating:** 6
**Confidence:** 4

**Summary:**

This paper considers a high-dimensional contextual linear bandit setting and uses recent results on
the risk of minumum
norm linear interpolators to evaluate the regret of two variants of an explore-then-commit algorithm.
The first variant makes use of the risk results to balance regret in the "explore" vs "commit"
phases, assuming oracle knowledge of the population covariance of the contexts (which
determine the risk, hence regret, of the linear interpolator and derived arms). The second variant
uses simple sample-covariance-based estimators that do not require such oracle knowledge to inform
the stopping of the explore phase, making the algorithm adaptive to the instance.

**Strengths:**

The paper's ideas are straightforward, and relatively easy to understand once one is familiar
with the work analyzing the risk of interpolating linear estimators. The paper also lays
out the results and proof ideas well. The algorithms are
also easy to motivate from that context, and amenable to quick implementation. The setting
of high-dimensional linear contextual bandits without sparsity is also interesting, pointing towards potentially
infinite-dimensional contexts.

**Weaknesses:**

In my view, the papers main limitations are related to its reliance on
the interpolating estimator work:
1. It is unclear to me why the paper is focused on covariance structures of the type
in Proposition 1. Presumably this is because these are the covariances for which the
minimum norm interpolators have non-trivial risk guarantees, but that seems like
a weak reason to focus solely on them; e.g. one could consider ridge estimators, lasso
estimators (notwithstanding the lack of sparsity, given it can still have good prediction error), etc.
2. The authors provide a lower bound for explore then commit algorithms showing that
their upper bound is tight, restricted to this class of algorithms. However, it is not obvious
that ucb or thompson sampling algorithms would face the same obstacle in performance. An information-theoretic
lower bound would be convincing on this score.

**Questions:**

1. Can you relate/situate this work with e.g. the paper "Model selection for Contextual Bandits" by Foster et al. and the references therein?
The settings are similar, but different in crucial ways (e.g. in their work context covariances are assumed bounded below).
2. Can you elaborate on the regret dependence in terms of $\theta_\max$? Presumably knowledge of this affects the optimal choice of $T_0$?
3. Can the class of covariance structures considered "benign" be expanded if one does not use interpolating estimators?
4. Can the example of Theorem 5 provide a "hard" example for more than explore-then-commit algorithms?

**Limitations:**

See above

---

> ### Author Rebuttal · Authors · 2023-08-10
>
> Thank you for your time and many comments.
>
> > It is unclear to me why the paper is focused on covariance structures of the type in Proposition 1. Presumably this is because these are the covariances for which the minimum norm interpolators have non-trivial risk guarantees, but that seems like a weak reason to focus solely on them; e.g. one could consider ridge estimators, lasso estimators (notwithstanding the lack of sparsity, given it can still have good prediction error), etc.
>
> Your point is spot-on. We are interested in this covariance structure because we can show that the ridge-less least-square estimator can yield a consistent estimator under this covariance structure. Regarding the ridge and lasso estimators, under the same covariance, non-sparse high-dimension settings these estimators are inconsistent (e.g., Tsigler & Bartlett (2020), Dobrian & Wagner (2019)). Hence, the ridge/lasso estimator is not suitable in this setting.
>
> > The authors provide a lower bound for explore then commit algorithms showing that their upper bound is tight, restricted to this class of algorithms. However, it is not obvious that ucb or thompson sampling algorithms would face the same obstacle in performance. An information-theoretic lower bound would be convincing on this score.
>
> You are right, our lower bound is for the class of explore-then-commit algorithms.
>
> > Can you relate/situate this work with e.g. the paper "Model selection for Contextual Bandits" by Foster et al. and the references therein? The settings are similar, but different in crucial ways (e.g. in their work context covariances are assumed bounded below).
>
> Foster et al. considered a linear bandit problem where the hypothesis space is nested with dimensions $d_1 < d_2 < d_3 < \dots < d_M$. Our AEtC and Foster et al. are in common in the sense that they both find the best sample complexity in linear contextual bandit problem. However, their sample complexity is determined to deal with finite dimension problem with $d_m$ (after a known map $\phi(x, a)$), whereas we consider $p$ non-zero dimensions with $p \ge T$ or even infinite without any such a map.  We will revise the paper to add this.
>
> > Can you elaborate on the regret dependence in terms of $\theta_{\mathrm{max}}$? Presumably knowledge of this affects the optimal choice of  $T_0$?
>
> Thank you for your question. Regret depends on linearly on $\theta_{\mathrm{max}}$, which we assumed to be a constant. If we consider this to be a variable, it matters to the choice of $T_0$.
>
> > Can the class of covariance structures considered "benign" be expanded if one does not use interpolating estimators?
>
> We can consider Ridge and Lasso regressors, but these regressors do not have consistency in our setting (Tsigler & Bartlett (2020), Dobrian & Wagner (2019)). Therefore, the choice of OLS is indispensable.
>
> > Can the example of Theorem 5 provide a "hard" example for more than explore-then-commit algorithms?
>
> For UCB or more adaptive algorithms where the sample depends on the history, existing results on benign overfitting do not apply because these theories are on a fixed number of samples (i.e., $N_i(t)$ cannot change adaptively).
>
> [references]
> *  Tsigler and  Bartlett (2023). Benign overfitting in ridge regression. JMLR.
> *  Dobriban, E., & Wager, S. (2018). High-dimensional asymptotics of prediction: Ridge regression and classification. AoS.

---

### Official Review · Reviewer_dtiS · 2023-07-07

**Soundness:** 2 fair
**Presentation:** 2 fair
**Contribution:** 2 fair
**Rating:** 5
**Confidence:** 2

**Summary:**

The authors studied the high-dimensional linear contextual bandit problem where the number of features 𝑝 is greater than the budget 𝑇, or it may even be infinite.

Problem setting: They assumed each arm has an individual hidden parameter, and the contexts are drawn from nature for each round.

They argue that only by using the notion of effective dimension, in some cases when data distributions have a small effective rank, explore-then-commit algorithm can address this problem.

**Strengths:**

They adopted the new techniques from [Bartlett et al., 2020] to design an explore-then-commit algorithm with this 'tough' environment - where the dimension $p$ is large compared to the number of samples. Seems like this observation is original.

Plus, they proved that their regret bound is tight by proving several lower bounds. If their arguments are all correct, these results can be significant.

**Weaknesses:**

I am unsure whether this paper is clear enough.

1) It's not straightforward whether they assumed 'benign covariance matrix' in proposition 1 or their assumption 1 implies benign covariance matrix.

2) I cannot understand how one can take an inverse matrix in line 117 when $p>n$.

3) Eventually, seems like they made only two major assumptions - Assumption 1 with sub-gaussianity, and Assumption 2 which is about the error rate of the OLS. Especially, for AEtC, they only used Assumption 1. I am strongly doubtful that they achieve a meaningful regret bound without any additional assumption on high-dimensional space - As in the first paragraph of Chapter 7, 'High-dimensional statistics' by Martin Wainwright, if the model lacks any additional structure, then there is no hope of getting a consistent estimator in the high-dimensional environment.

**Questions:**

0) In line 117, you mentioned that $\hat{\theta}$ can be achieved by simple OLS (ordinary least square) operation. However, you mentioned that $p>n$. How can you define $(XX^\top)^{-1}$ in this case? The authors also mentioned that their analysis can be executable even when $p=\infty$ (the first sentence of the abstract.) Please help me to understand this.


1) The authors use the word 'optimal balance' when they describe AEtC. Here's the sentence:

> Moreover, we introduce an adaptive explore-then-commit (AEtC) algorithm that adaptively finds the optimal balance.

I want to ask what the word 'optimal' means. I cannot see any lower-bound proof of the exploration in this paper. Did you prove that none of the EtC algorithms can beat the AEtC algorithm? If not, I believe that one should be careful of using the word 'optimal.'


2) According to my understanding, AEtC is an algorithm that relieves the assumption on the knowledge of $\Sigma^{(i)}$ (and naturally Assumption 2) from EtC, not the algorithm that improves the order of regret, am I right?


3) I want to know whether you 'assumed' the covariance matrix to be benign, or Assumption 1 (or 2) implies the covariance matrix to be benign.


4) In your proposition 1, I want to check that your $\lambda_k^{(i)}$ is exactly $k^{-1 + 1/T^a}$ or $k^b$, not the orderwise notation, am I right?


5) Maybe a silly question, but suppose that we regard your problem as a K-armed bandit problem, without considering the context. Then, isn't this mean that we already have a regret bound of $O(\sqrt{KT})$? Even for EtC, we have $O(T^{2/3})$ regret bound for K-armed bandit with EtC. What is the main factor that separates your problem from the naive K-armed bandit problem?

**Limitations:**

There's no negative societal impact of their work.
I also think they didn't state their limitations or future works properly.

---

> ### Author Rebuttal · Authors · 2023-08-10
>
> We thank your careful reading and insightful comments. The score is a bit low compared with the tone of the review. The main questions raised by the reviewers are about the problem setting. To address this, we will revise the paper as outlined in the global reply. Below, we answer each of the questions.
>
> > It's not straightforward whether they assumed 'benign covariance matrix' in proposition 1 or their assumption 1 implies benign covariance matrix.
>
> Assumption 1 shows that the elements of this random matrix are independent sub-Gaussian (light tail probability). It is possible to have benign covariance (Definition 1) but not satisfy Assumption 1, and it is possible to satisfy Assumption 1 but not have benign covariance. I will add an explanation for this.
>
> > I cannot understand how one can take an inverse matrix in line 117 when $p > n$.
> > In line 117, you mentioned that can be achieved by simple OLS (ordinary least square) operation. However, you mentioned that. How can you define in this case?
>
> We consider situations where $p>n$.
> Our estimator takes the inverse of the matrix $X X^T$, so we have $n \times n$ matrices with rank $n$, and the estimator $X^T (X X^T)^{-1} Y$ (e.g., line 117) does not cause a problem.
> Note that this estimator can be rewritten as $ (X^T X)^{-1} X^T Y$, in which case it takes the inverse of $X^T X$. Since this matrix is $p \times p$ and the rank is $n$, we cannot take the inverse matrix, but we can take the generalized inverse matrix.
>
>
>
> > Eventually, seems like they made only two major assumptions - Assumption 1 with sub-gaussianity, and Assumption 2 which is about the error rate of the OLS. Especially, for AEtC, they only used Assumption 1. I am strongly doubtful that they achieve a meaningful regret bound without any additional assumption on high-dimensional space.
>
> We make specific assumptions on the eigenvalue structure of the covariance matrix $\Sigma$. Namely, the assumption of benign covariance as given in Definition 1. Under this assumption, the benign overfitting theory proves that we can achieve a consistent estimator in high dimensions (Bartlett et al. (2020)).
>
> > Meaning of “optimal” here. “Moreover, we introduce an adaptive explore-then-commit (AEtC) algorithm that adaptively finds the optimal balance.”
> > According to my understanding, AEtC is an algorithm that relieves the assumption on the knowledge of  (and naturally Assumption 2) from EtC, not the algorithm that improves the order of regret, am I right?
>
> The optimality here is on the choice of $T_0$. To optimize it, EtC needs to know the spectrum ${\lambda_k}$, and AEtC adaptively estimates it. Thank you for the clarification.
>
> > I want to know whether you 'assumed' the covariance matrix to be benign, or Assumption 1 (or 2) implies the covariance matrix to be benign.
>
> Assumption 2 given in EtC's theorem (assumed to be of error order) is necessary for the covariance matrix to be benign (Definition 1).
>
> Theorem 9 for AEtC more directly assumes the eigenvalue structure of the covariance matrix (see the sentence “Suppose DGP of Example 1 or Example 2 in Proposition 1.” in the statement). This makes the covariance matrix benign, allowing the matching estimator to be constructed naturally.
>
>
> > In your proposition 1, I want to check that your $\lambda_k^{(i)}$ is exactly $k^{-1+1/T^a}$ or $k^b$, not the orderwise notation, am I right?
>
> It is exactly $k^{-1+1/T^a}$ or $k^b$, but multiplying a constant $C$ to all eigenvalues is easy.
>
> > Maybe a silly question, but suppose that we regard your problem as a K-armed bandit problem, without considering the context. Then, isn't this mean that we already have a regret bound of $O(\sqrt{KT})$? Even for EtC, we have regret bound $O(T^{2/3})$ for K-armed bandit with EtC.
>
> As you mentioned, our problem is contextual (that means, to have sublinear regret, we must choose the best arm $i^*(t)$ for each round $t$ (Eq. (2)). Therefore the regret here is much more challenging than the regret of context-free K-armed bandit problem (where the best arm is consistent) and results on context-free K-armed bandit problem do not apply.
>
> [references]
> * Bartlett et al. (2020). Benign overfitting in linear regression. PNAS.
> * Tsigler and  Bartlett (2023). Benign overfitting in ridge regression. JMLR.

---

> > ### Author Response · Authors · 2023-08-14
> > **Thank you for your time & our update plan**
> >
> > Dear reviewer,
> >
> > We will make significant changes to the introduction of the setting as described in the global reply. If you still have concerns about the setting, we are happy to address them. Thank you for your time and efforts.

---

> > > ### Comment · Reviewer_dtiS · 2023-08-21
> > > **Thanks for your response.**
> > >
> > > Thanks for your clear response. I just changed my score from 4 to 5.

---

> > > > ### Author Response · Authors · 2023-08-21
> > > > **Thank you for all questions**
> > > >
> > > > Thank you for taking the time to review this paper! If your assessment aligns with the paper being above the acceptance threshold, would you consider increasing the score to weak accept that reflects the actual decision threshold? I understand that this comes toward the end of the process, and making changes may not be comfortable. If you prefer to keep the current score, we completely understand.
> > > >
> > > > Your inquiries, particularly those related to the clarity of the assumptions and the robustness of our findings, are greatly valued. They are helpful in our future revisions.

---

### Official Review · Reviewer_pLAf · 2023-07-07

**Soundness:** 3 good
**Presentation:** 2 fair
**Contribution:** 3 good
**Rating:** 6
**Confidence:** 3

**Summary:**

This paper addresses the linear contextual bandit problem in high dimensions where the number of features is greater than the budget.  While the recent works impose an assumption about sparsity on the regression coefficients, this paper uses the minimum-norm interpolating estimator which is inspired by recent results on overparameterized models when data distributions have small effective ranks. The authors propose an explore-then-commit algorithm and derive an optimal regret for the proposed algorithm under their setting. This algorithm requires model-dependent parameters on the covariance, which can limit the applications in practice. Therefore, they further propose
an adaptive explore-then-commit algorithm by estimating these parameters and show that their adaptive version also achieves the optimal rate. Finally,  they provide some simulations to verify the efficiency of the proposed method.

**Strengths:**

- Unlike previous works, the paper solves the high-dimensional linear contextual bandit problem without sparsity. This seems to be the first work in the field of high-dimensional contextual bandits. This paper also proposes new algorithms in the form of the Explore-then-Commit strategy in their setting and derives the optimal regrets for proposed algorithms.

- The organization and the presentation are almost clear to follow.



**Weaknesses:**

- Although the paper introduces a new direction for the high-dimensional linear contextual bandits, the authors seem not to analyze deeply the difference between their work and the recent works using sparsity (e.g., Kim and Paik, 2019, Bastani and Bayati, 2020, Hao et al., 2022). Without sparsity, their work needs to impose a condition to be benign on the covariance matrix as well as Assumption 2 in the paper. I am wondering if this assumption is weaker than the assumption on the sparsity of the recent works. Thus, I think that a section of the related works and a comparison of assumptions with recent works are needed.
- The regret bounds in Theorem 4 and Theorem 9 are in terms of $\alpha$. However, the definition of $\alpha$ is not clear at all. For examples 1 and 2 in the paper, $\alpha$ can be understood but in general, what is $\alpha$?
-  Although the regret bounds are provided for the proposed algorithms, I think that a comparison of regret bounds with the recent works (e.g., Hao et al., 2022) is also needed.
-  The used techniques are adapted from the recent work of Bartlett et al., 2020.



**Questions:**

- In Theorem 4, they impose Assumptions 1 and 2. Thus, where is the assumption about the benign covariance?
- Also see my questions in the Weaknesses section

**Limitations:**

Yes

---

> ### Author Rebuttal · Authors · 2023-08-10
>
> Thank you for your insightful review. These questions are fundamental to the benign assumption, and we will revise the paper based on your comments. Below, we answer the questions raised.
>
> > Although the paper introduces a new direction for the high-dimensional linear contextual bandits, the authors seem not to analyze deeply the difference between their work and the recent works using sparsity (e.g., Kim and Paik, 2019, Bastani and Bayati, 2020, Hao et al., 2022). Without sparsity, their work needs to impose a condition to be benign on the covariance matrix as well as Assumption 2 in the paper. I am wondering if this assumption is weaker than the assumption on the sparsity of the recent works. Thus, I think that a section of the related works and a comparison of assumptions with recent works are needed.
>
> Sparse and benign overfitting are orthogonal. Many examples are benign but non-sparse and vice versa. The main motivation behing benign overfitting theory is to explain the learnability of recent large-scale models. To put it differently, we consider different classes of problems where we can guarantee sublinear regret bound irrespective of sparseness assumption.
>
> > The regret bounds in Theorem 4 and Theorem 9 are in terms of $\alpha$. However, the definition of $\alpha$ is not clear at all. For examples 1 and 2 in the paper, can be understood but in general, what is $\alpha$?
>
> $T^{\alpha}$ is such that $N = T E(N, T)$ (Eq.(7) with $K=O(1)$), and its closed formula is derived in Examples 1 and 2. The $\alpha$ in Theorem 4 and 9 is identical. We thank the reviewer for pointing out this and will revise the paper to clarify this.
>
> > Although the regret bounds are provided for the proposed algorithms, I think that a comparison of regret bounds with the recent works (e.g., Hao et al., 2022) is also needed.
> The used techniques are adapted from the recent work of Bartlett et al., 2020.
>
> The complexity of a sparse bandit algorithm is characterized by a number of non-zero features $s$. In our case, $s$ corresponds to $p$ (i.e., all dimensions are non-zero), and for any $p \ge T$, all sparse bandit regret bounds are meaningless $\Omega(T)$ bounds (c.f., Li et al. Table 1, Jang et al. Table 1) unlike our algorithms that have sublinear regret for $p \ge T$. We will add this discussion in the revised version.
>
> > In Theorem 4, they impose Assumptions 1 and 2. Thus, where is the assumption about the benign covariance?
>
> Thank you for the important clarification question. Definition 1 is the heart of the benign property.
> Following (Bartlett et al 2020, Tsigler and Bartlett, 2020), Assumption 1 is used to derive our bound, but this assumption is not necessary for the benign property (i.e., there are benign examples without Assumption 1).
>
> Thank you for many other suggestions. We will update the paper accordingly.

---

> ### Comment · Reviewer_pLAf · 2023-08-21
>
> Thank you for your rebuttal. I have no further questions.

---

### Official Review · Reviewer_h4Ff · 2023-07-12

**Soundness:** 3 good
**Presentation:** 3 good
**Contribution:** 3 good
**Rating:** 6
**Confidence:** 3

**Summary:**

This paper analyzes the Explore-then-commit algorithm for the linear contextual bandit problem in high dimensions, when the number of features is higher than the number of rounds T. The authors use recent studies on overparametrized models for linear regression to analyze the algorithm. Since this algorithm requires the knowledge of the underlying covariance matrices of the contexts, the authors provide an alternate algorithm where they adaptively determine the extent of exploration required.

**Strengths:**

The paper presents a thorough analysis of the high-dimensional linear contextual bandit problem by providing upper and matching lower bounds on the estimation error. The Adaptive algorithm presented in the paper further adds to the paper's strengths and provides a practical way to solve the problem. The simulations provided demonstrate the the adaptive algorithm works practically in both the benign and non-benign cases.



**Weaknesses:**

The authors could improve the writing of the paper in the following ways:
1. The introduction of the term benign covariance is a bit abrupt and very technical very quickly. The authors could provide more intuition about what it means before defining the bias and variance terms.

2. The effective rank definitions used in this paper vary slightly from the standard definitions in literature. The authors could maybe visualize the spectra of a matrix and illustrate what these quantities correspond to.


**Questions:**

1. Could the authors define the conditional number of a matrix? If they are referring to the standard condition number, how can this number be negative?

---

> ### Author Rebuttal · Authors · 2023-08-10
>
> Thank you for your insightful review. We would improve the introduction of benign overfitting theory as suggested.
>
> > The introduction of the term benign covariance is a bit abrupt and very technical very quickly. The authors could provide more intuition about what it means before defining the bias and variance terms.
> > The effective rank definitions used in this paper vary slightly from the standard definitions in literature.
>
> This terminology derives from the theory of benign overfitting (Bartlett et al 2020, Tsigler and Bartlett, 2020). We will add a more detailed discussion of these terminologies. Please see the global reply for its outline. Effective rank characterizes the complexity of the problem where the spectrum is considered. Eigenvalues that are close to zero make little contribution to the effective rank. Coherent rank splits eigenvalues into major and minor ones. Bias corresponds to loss due to the minor eigenvalues that are unlearnable but acceptable. Variance corresponds to the loss from the major eigenvalues that are learnable.
>
>
> > The authors could maybe visualize the spectra of a matrix and illustrate what these quantities correspond to.
>
> Example 1 states that the spectrum decays slightly faster than $k^{-1}$. Example 2 states that spectrum decays slower than $k^{-1}$ but $p$ is finite. The simulation also involves exponential decay. We will add a visualized spectrum of these settings in the camera-ready version. Please see the pdf for figures in the global reply.
>
> Note also that, in the context of benign overfitting, our main innovations in view of the overparameterized theory are as follows:
> Unlike (Bartlett et al 2020, Tsigler and Bartlett, 2020), we consider the case the number of data points used in training ($N$ in our paper) is smaller than full data of size $T$. We introduce the method for estimating the optimal rate. The utility of this result is not limited to bandit setting but can benefit the overparameterization theory.
>
>
> > Could the authors define the conditional number of a matrix? If they are referring to the standard condition number, how can this number be negative?
>
> We define the conditional number as the ratio between the largest and the smallest singular values, which is non-negative but can be zero. Theorem 2 excludes some corner cases where the conditional number is zero with non-negligible probability.
>
> [references]
> * Bartlett et al. (2020). Benign overfitting in linear regression. PNAS.
> * Tsigler and  Bartlett (2023). Benign overfitting in ridge regression. JMLR.

---

> > ### Comment · Reviewer_h4Ff · 2023-08-18
> >
> > Thank you for the clarifications! I do not have any further comments.

---

### Author Rebuttal · Authors · 2023-08-10

Thank you all for your thoughtful and important comments.
We are making the following major changes based on your comments.

> Add more introduction to benign covariance and related notion

To enhance our readers' comprehension of overparameterized models, we propose to provide a more comprehensive introduction to this topic. In particular, we feel it would be beneficial to emphasize that many of our underpinning assumptions align seamlessly with the established norms in the field (Bartlett et al. (2020) and Tsigler and Bartlett (2020)).

When the eigenvalues decay at a certain rate, the error convergences to zero even in high or infinite dimensions. An effective rank is introduced to more rigorously capture the decay of eigenvalues. This notion has been used in a random analysis for matrices (e.g. Koltchinskii et al. (2017)). We will add new figures to describe how the eigenvalue decay affects the effective rank.

We append a plot of effective rank $r_k(\Sigma), R_k(\Sigma)$ up to dimension $k$, in order to give intuitions on the effective rank. Even though we consider full-rank covariance matrices, effective ranks can be sublinear to the number of dimensions.

> Clarify our innovation in the theory of benign covariance

Based on the overparameterized theory, our main innovations in the context of the overparameterized theory are as follows:
* Unlike the standard overparameterized theory (Bartlett et al 2020, Tsigler and Bartlett, 2020), we consider the case the number of data points used in the training ($N$ in our paper) is smaller than full data of size $T$.
* We introduce the method for estimating the optimal rate. The utility of this result is not limited to bandit setting but can benefit the overparameterization theory.

> Add discussion on comparison with bandit with sparse context

The complexity of a sparse bandit algorithm is characterized by the number of non-zero features $s$. In our case, $s$ corresponds to $p$ (i.e., all dimensions are non-zero), and for any $p \ge T$ all sparse bandit regret bounds are meaningless $\Omega(T)$ bounds (c.f., Li et al. Table 1, Jang et al. Table 1) unlike our algorithms that have sublinear regret for $p \ge T$. We will add this discussion in the camera ready.

[references]
* Bartlett et al. (2020). Benign overfitting in linear regression. PNAS.
*  Tsigler and  Bartlett (2023). Benign overfitting in ridge regression. JMLR.
* Jang et al. (2022) PopArt: Efficient Sparse Regression and Experimental Design for Optimal Sparse Linear Bandits, NeurIPS.
* Li et al. (2022) A simple unified framework for high dimensional bandit problems, ICML.
* Koltchinskii & Lounici,  (2017). Concentration inequalities and moment bounds for sample covariance operators. Bernoulli.

---

### Decision · Program_Chairs · 2023-09-21

**Decision:**

Accept (poster)

**Comment:**

The reviewers all agree that this paper studies overparameterized models for linear regression for bandits, introducing this new problem setup to the bandit community.

The reviewers requested that the authors add more explanations in the final version, as the original submission is technically too fast for most readers and the use of tools are not commonly known to the bandit community.